

# Rossby number similarity of atmospheric RANS using limited length scale turbulence closures extended to unstable stratification

Maarten Paul van der Laan[1], Mark Kelly[1], Rogier Floors[1], and Alfredo Peña[1]

[1]Technical University of Denmark, DTU Wind Energy, Risø Campus, Frederiksborgvej 399, 4000 Roskilde, Denmark

**Correspondence:** Maarten Paul van der Laan (plaa@dtu.dk)

**Abstract.** The design of wind turbines and wind farms can be improved by increasing the accuracy of the inflow models representing the atmospheric boundary layer. In this work we employ one-dimensional Reynolds-averaged Navier-Stokes (RANS) simulations of the idealized atmospheric boundary layer (ABL), using turbulence closures with a length scale limiter. These models can represent the mean effects of surface roughness, Coriolis force, limited ABL depth, and neutral and stable atmo-
spheric conditions using four input parameters: the roughness length, the Coriolis parameter, a maximum turbulence length, and the geostrophic wind speed. We find a new model-based Rossby similarity, which reduces the four input parameters to two Rossby numbers with different length scales. In addition, we extend the limited length scale turbulence models to treat the mean effect of unstable stratification in steady-state simulations. The original and extended turbulence models are compared with historical measurements of meteorological quantities and profiles of the atmospheric boundary layer for different atmospheric
stabilities.

## 1 Introduction

Wind turbines operate in the turbulent atmospheric boundary layer (ABL) but are designed with simplified inflow conditions that represent analytic wind profiles of the atmospheric surface layer (ASL). The ASL corresponds to roughly the first 10% of ABL, typically less than 100 m, while the tip height of modern wind turbines are now sometimes beyond 200 m. Hence, there
is a need for inflow models that represent the entire ABL in order to improve the design of wind turbines and wind farms. Such a model should be simple enough to be applicable in the wind energy industry.

The ABL is complex and changes continuously over time. Idealized, steady-state models can represent long-term averaged velocity and turbulence profiles of the real ABL, including effects of Coriolis, atmospheric stability, capping inversion, a homogeneous surface roughness and flat terrain. In this work, we investigate idealized ABL models that are based on one-
dimensional Reynolds-averaged Navier-Stokes (RANS), where the only spatial dimension is the height above ground. The output of the model can be used as inflow conditions for three-dimensional RANS simulations of complex terrain (Koblitz et al., 2015) and wind farms (van der Laan and Sørensen, 2017b). Turbulence is here modeled by two limited length scale turbulence closures, the mixing-length model of Blackadar (1962) and the two equation $k$-$\varepsilon$ model of Apsley and Castro (1997). These turbulence models can simulate stable and neutral ABLs without the necessity of a temperature equation and a momentum
source term of buoyancy. In other words, all temperature effects are represented by the turbulence model. The limited length



scale turbulence models depend on four parameters: the roughness length, the Coriolis parameter, the geostrophic wind speed and a chosen maximum turbulence length scale that is related to the ABL depth. We show that the normalized profiles of wind speed, wind direction and turbulence quantities are only dependent on two dimensionless parameters that represent the ratio of the inertial to the Coriolis force, based on two different length scales; the roughness length and the maximum turbulence length

scale. These dimensionless parameters are Rossby numbers. The Rossby number based on the roughness length is known as the surface Rossby number as introduced by Lettau (1959), while the Rossby number based on the maximum turbulence length is a new dimensionless parameter. The obtained model-based Rossby number similarity is used to validate a range of simulations with historical measurements of geostrophic drag coefficient and cross isobar angle. In addition, we show that both RANS models' solutions are bounded by two analytic solutions of the idealized ABL.

The limited length scale turbulence closures of Blackadar (1962) and Apsley and Castro (1997) can model the effect of stable and neutral stability but cannot model the unstable atmosphere. We propose simple extensions to solve this issue and validate the results of the extended $k$-$\varepsilon$ model with measurements of wind speed and wind direction profiles. The model extensions lead to a third Rossby number, where the length scale is based on the Obukhov length. The limited mixing-length model is not considered in the comparison with measurements because we are mainly interested in the limited length scale $k$-$\varepsilon$ model. The

$k$-$\varepsilon$ model is more applicable to wind energy applications because it can also provide an estimate of the turbulence intensity, which is not available from limited mixing-length (Blackadar type) models. The limited mixing-length model is applied here to show that the same model-based Rossby number similarity is recovered as obtained for the $k$-$\varepsilon$ model.

The article is structured as a follows. Background and theory of the idealized ABL are discussed in Section 2. Extensions to unstable surface layer stratification are presented in Section 3. Section 4 presents the methodology of the one dimensional

RANS simulations. The model-based Rossby similarity is illustrated in Section 5. The simulation results of the limited length scale $k$-$\varepsilon$ model including the extension to unstable conditions are compared with measurements in Section 6.

## 2  Background and theory: idealized models of the ABL

We model the mean steady-state flow in an idealized ABL. Here idealized refers to flow over homogeneous and flat terrain under barotropic conditions such that the geostrophic wind does not vary with height. This flow can be described by the

incompressible RANS equations for momentum, where the contribution from the molecular viscosity is neglected due to the high Reynolds number:

$$\frac{DU}{Dt} = f_c(V - V_G) - \frac{d}{dz}\left(\nu_T \frac{dU}{dz}\right) = 0, \qquad \frac{DV}{Dt} = -f_c(U - U_G) - \frac{d}{dz}\left(\nu_T \frac{dV}{dz}\right) = 0, \tag{1}$$

where $U$ and $V$ are the mean horizontal velocity components, $U_G$ and $V_G$ are the corresponding mean geostrophic velocities, $f_c = 2\Omega \sin(\lambda)$ is the Coriolis parameter with $\Omega$ as Earth's angular velocity and $\lambda$ as the latitude, and $z$ is the height above

ground. In addition, the Reynolds-stresses $\overline{u'w'}$ and $\overline{v'w'}$ are modeled by the linear stress-strain relationship of Boussinesq (1897): $\overline{u'w'} = \nu_T dU/dz$ and $\overline{v'w'} = \nu_T dV/dz$, where $\nu_T$ is the eddy viscosity. The boundary conditions for $U$ and $V$ are: $U = V = 0$ at $z = z_0$ and $U = U_G$ and $V = V_G$ at $z \to \infty$, where $z_0$ is the roughness length. Note that it is possible to write



the two momentum equations as a single ordinary differential equation:

$$\frac{d}{dz}\left(\nu_T \frac{dW}{dz}\right) = i f_c W, \tag{2}$$

where $W \equiv (U - U_G) + i(V - V_G)$ is a complex variable and $i^2 = -1$.

The eddy viscosity, $\nu_T$, needs to be modeled in order to close the system of equations. The eddy viscosity can be written
as $\nu_T = u_* \ell$, where $u_*$ and $\ell$ represent turbulence velocity and turbulence length scales. For a constant eddy viscosity, the
equations can be solved analytically and the solution is known as the Ekman spiral (Ekman, 1905), which includes the wind
direction change with height due to Coriolis effects. The Ekman spiral can also be considered a laminar solution, since one
can neglect the turbulence in the momentum equations and set the molecular viscosity to determine the rate of mixing. For an
eddy viscosity that increases linearly with height, the equations can also be solved analytically, as introduced by Ellison (1956)
and discussed by Krishna (1980) and Constantin and Johnson (2019). The two analytic solutions are provided in Appendix A.
One can relate the analytic solution of Ellison (1956) to the (neutral) ASL ($z \ll z_i$), while the Ekman spiral is more valid
for altitudes around the ABL depth $z_i$. Neither of the two analytic solutions result in a realistic representation of the entire
(idealized) ABL. A combination of both a linear eddy viscosity for $z \ll z_i$ and a constant eddy viscosity for $z \sim z_i$ should
provide a more realistic solution. For example, the eddy viscosity could have the form $\nu_T = \kappa u_{*0} z \exp(-z/h)$, where $\nu_T$
increases linearly with height for $z \ll h$ as expected in the surface layer, then it reaches a maximum value at $z = h$, and
decreases to zero for $z > h$. Note that $u_{*0}$ is the friction velocity near the surface. Constantin and Johnson (2019) derived a
number of solutions for a variable eddy viscosity, although an explicit solution for the entire idealized ABL with a realistic
eddy viscosity (in the previously mentioned form) has not been found yet. Hence, numerical methods are still necessary, and
one of the simplest numerical model for the idealized ABL is given by Blackadar (1962) using Prandtl's mixing-length model:

$$\nu_T = \ell^2 \mathcal{S} \tag{3}$$

where $\mathcal{S} = \sqrt{(dU/dz)^2 + (dV/dz)^2} = |dW/dz|$ is the magnitude of the strain-rate tensor, and prescribed $\ell$ as a turbulence
length scale

$$\ell = \frac{\kappa z}{1 + \frac{\kappa z}{\ell_{\max}}}, \tag{4}$$

where $\kappa z$ is the turbulence length scale in the neutral surface layer with $\kappa$ as the von Kármán constant, and $\ell_{\max}$ is a maximum
turbulence length scale. It is also possible to model the eddy viscosity with a two-equation turbulence closure, e.g., the $k$-$\varepsilon$
model:

$$\nu_T = C_\mu \frac{k^2}{\varepsilon} \tag{5}$$

with $C_\mu$ as a model parameter, $k$ as the turbulent kinetic energy and $\varepsilon$ as the dissipation of $k$. Both $k$ and $\varepsilon$ are modeled by a
transport equation:

$$\frac{Dk}{Dt} = \frac{d}{dz}\left(\frac{\nu_T}{\sigma_k}\frac{dk}{dz}\right) + \mathcal{P} - \varepsilon, \tag{6}$$

$$\frac{D\varepsilon}{Dt} = \frac{d}{dz}\left(\frac{\nu_T}{\sigma_\varepsilon}\frac{d\varepsilon}{dz}\right) + (C_{\varepsilon,1}\mathcal{P} - C_{\varepsilon,2}\varepsilon)\frac{\varepsilon}{k}, \tag{7}$$





where $\mathcal{P}$ is the turbulence production, and $\sigma_k$, $\sigma_\varepsilon$, $C_{\varepsilon,1}$ and $C_{\varepsilon,2}$ are model constants that should follow the relationship $\kappa^2 = \sigma_\varepsilon \sqrt{C_\mu}(C_{\varepsilon,2} - C_{\varepsilon,1})$. When using the standard $k$-$\varepsilon$ model calibrated for atmospheric flows (Richards and Hoxey, 1993), the turbulence length scale or eddy-viscosity will keep increasing until a boundary layer depth is formed and the analytic solution of Ellison (1956) is approximated. Apsley and Castro (1997) proposed to modify the transport equation of $\varepsilon$, such that

a maximum turbulence length scale is enforced by replacing the constant $C_{\varepsilon,1}$ with a variable parameter $C^*_{\varepsilon,1}$:

$$C^*_{\varepsilon,1} = C_{\varepsilon,1} + (C_{\varepsilon,2} - C_{\varepsilon,1}) \frac{\ell}{\ell_{\max}}, \tag{8}$$

where the turbulence length scale is modeled as $\ell = C_\mu^{3/4} k^{3/2}/\varepsilon$. This limited-length scale $k$-$\varepsilon$ model behaves very similar to the mixing-length model of Blackadar (1962) (Eqs. 3 and 4). For $\ell \ll \ell_{\max}$, the surface layer solution is obtained, while for $\ell \sim \ell_{\max}$, the source terms in the transport equation of $\varepsilon$ cancel ($C^*_{\varepsilon,1}\mathcal{P} \sim C_{\varepsilon,1}\varepsilon$), and the turbulence length scale is limited.

For a given $z_0$, $G$, and $f_c$, the ABL depth can be controlled by $\ell_{\max}$. This means that $\ell_{\max}$ is related to $z_i$; Apsley and Castro (1997) noted that $\ell_{\max} \sim z_i/3$ for typical neutral ABLs. However, the simulated boundary layer depth using the $k$-$\varepsilon$ model of Apsley and Castro (1997) has an approximate dependence of $z_i \propto (G/|f_c|)^{1-a}\ell_{\max}^a$ with $a \approx 0.6$, which we will further discuss in Section 5. A summary of the discussed eddy viscosity closures is listed in Table 1. Figure 1 compares the analytic solutions

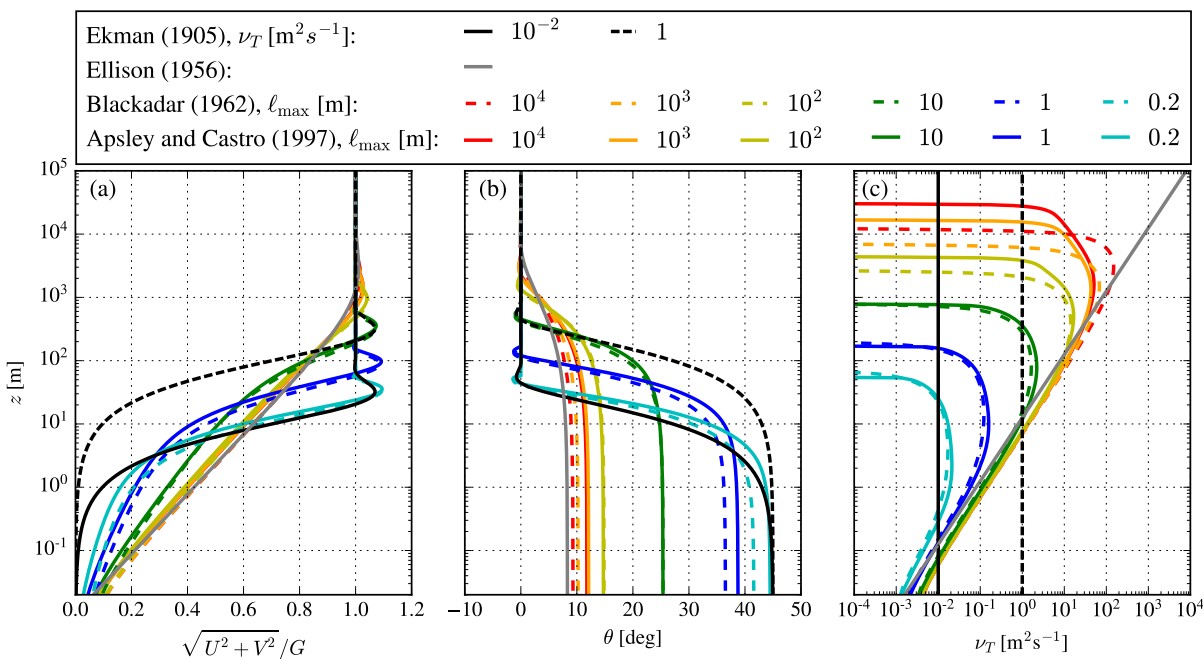

**Figure 1.** Comparison of analytic and numerical solutions of existing idealized ABL models using $f_c = 10^{-4}$ s$^{-1}$, $G = 10$ m s$^{-1}$ and $z_0 = 10^{-2}$ m for different model parameters. **(a)** Wind speed. **(b)** Wind direction. **(c)** Eddy viscosity.

of Ekman (1905) and Ellison (1956) with the numerical solutions of the limited mixing-length model of Blackadar (1962) and

the limited length scale $k$-$\varepsilon$ of Apsley and Castro (1997) in terms of wind speed, wind direction, $\theta = \arctan(V/U)$, and eddy



viscosity. The Ekman spiral is depicted with two constant eddy viscosities, which only translates the solution vertically. In addition, we have chosen $f_c = 10^{-4}$ s$^{-1}$, $G = 10$ ms$^{-1}$, and $z_0 = 10^{-2}$ m. The numerical solutions are shown for a range of $\ell_{max}$ values. It is clear that the ABL depth decreases for lower values of $\ell_{max}$, for both numerical models, and their solutions behave similarly. A lower $\ell_{max}$ also results in a higher shear and wind veer, and a lower eddy viscosity, which are characteristics

of a stable ABL. Hence, the limited length scale turbulence closures can model the effects of stable stratification by solely limiting the turbulence length scale, without the need of a temperature equation or buoyancy source terms. When $\ell_{max} \to 0$ m (note that there is minimal limit of $\ell_{max}$ in order to obtain numerically stable results), the solution approaches to the Ekman spiral because the eddy viscosity in the ABL can be approximated by a constant eddy viscosity. Hence, the maximum change in wind direction simulated by the $k$-$\varepsilon$ model of Apsley and Castro (1997) is that of the Ekman spiral: 45°. For large $\ell_{max}$

values, the numerical solution approximates the analytic solution of Ellison (1956) but does not match it because their eddy viscosities are different for $z \geq z_i$.

| Eddy viscosity closure | | | Solution | Reference |
|---|---|---|---|---|
| Constant | - | - | Analytic | Ekman (1905) |
| Linear | $\nu_T = u_{*0}\ell$ | $\ell = \kappa z$ | Analytic | Ellison (1956) |
| Limited mixing-length model | $\nu_T = \ell^2 \mathcal{S}$ | $\ell = \kappa z/(1 + \kappa z/\ell_{max})$, | Numerical | Blackadar (1962) |
| Limited length scale $k$-$\varepsilon$ model | $\nu_T = C_\mu k^2/\varepsilon$ | $\ell = C_\mu^{3/4} k^{3/2}/\varepsilon$ | Numerical | Apsley and Castro (1997) |

**Table 1.** Eddy viscosity closures for the idealized ABL.

## 3   Extension to unstable surface layer stratification

The two limited length scale turbulence closures discussed in Section 2 can be used to model neutral and stable ABLs without the need of a temperature equation and buoyancy forces. However, it is not possible to model the unstable ABL because the

turbulence length scale is only limited, not enhanced, i.e., $\ell \leq \kappa z$. In order to model unstable conditions, we need to extend the models such that the turbulence length scale is enhanced in the surface layer, $\ell > \kappa z$, which we present in the following sections for each turbulence closure.

### 3.1   Limited mixing-length model

One can generically parameterize the turbulence length scale $\ell$ as a 'parallel' combination of ASL and ABL scales,

$$\frac{1}{\ell} = \frac{1}{\ell_{ASL}} + \frac{1}{\ell_{ABL}} \qquad (9)$$

Blackadar (1962) chose $\ell_{ASL} = \kappa z$ and $\ell_{ABL} = \ell_{max}$ to arrive at Eq. (4). If we choose to set

$$\ell_{ASL} = \frac{\kappa z}{\phi_m} \qquad (10)$$



following the turbulence length scale that is a result of Monin-Obukhov Similarity Theory (MOST, Monin and Obukhov (1954)) —where

$$\phi_m = (1 - \gamma_1 z/L)^{-1/4} \tag{11}$$

is the dimensionless velocity gradient for unstable conditions, with $\gamma_1 \approx 16$ as shown by Dyer (1974), and $L$ is the Obukhov

length—then it is possible to extend the limited mixing-length model of Blackadar (1962) to unstable surface layer stratification, as

$$\ell = \frac{\kappa z}{(1 - \gamma_1 z/L)^{-1/4} + \kappa z/\ell_{\max}}. \tag{12}$$

Approaching neutral conditions, $L^{-1} \to 0$ m$^{-1}$, the original length scale model of Blackadar (1962) is obtained. In stable conditions, $\phi_m = 1 + \beta z/L$, so the resulting turbulence length can also be rewritten in the Blackadar-type forms Eqs. (4) and

(9), using an effective maximum turbulence length scale of

$$\ell_{\text{ABL,stable}}^{-1} = \ell_{\max,\text{eff}}^{-1} \equiv \ell_{\max}^{-1} + \beta/(\kappa L). \tag{13}$$

### 3.2 Limited length scale $k$-$\varepsilon$ model

Sumner and Masson (2012) argued that for stable conditions, the limited length-scale $k$-$\varepsilon$ model of Apsley and Castro (1997) overpredicts $\ell$ in the surface layer compared to MOST, where $\ell_{\max} = L\kappa/\beta$ and $\beta \approx 5$. They proposed a more complicated

expression for $C_{\varepsilon,1}^*$ in the transport equation of $\varepsilon$ compared to the original model of Apsley and Castro (1997). Sogachev et al. (2012) alternatively prescribed a coefficient in the buoyant term of the $\varepsilon$ equation, depending on $\ell/\ell_{\max}$ and being similar to the production-related term that gives results consistent (at least asymptotically) with MOST. We find that the correction of Sumner and Masson (2012) provides a better match of the turbulence length scale within the surface layer compared to MOST with respect to the original $k$-$\varepsilon$ model of Apsley and Castro (1997). However, we also find that a larger overshoot of

the turbulence length scale around the ABL depth is found when Coriolis is included. Alternatively, one could improve the surface layer solution of the original model of Apsley and Castro (1997) by simply reducing $\ell_{\max}$ by roughly 20%. Therefore, we choose to use the model of Apsley and Castro (1997) as our starting point.

In order to account for the increase in turbulence length scale in the surface layer under unstable conditions, we add a buoyancy source term $B$ in the $k$-$\varepsilon$ transport equations:

$$\frac{Dk}{Dt} = \frac{d}{dz}\left(\frac{\nu_T}{\sigma_k}\frac{dk}{dz}\right) + \mathcal{P} - \varepsilon + B \tag{14}$$

$$\frac{D\varepsilon}{Dt} = \frac{d}{dz}\left(\frac{\nu_T}{\sigma_\varepsilon}\frac{d\varepsilon}{dz}\right) + \left(C_{\varepsilon,1}^*\mathcal{P} - C_{\varepsilon,2}\varepsilon + C_{\varepsilon,3}^*B\right)\frac{\varepsilon}{k}. \tag{15}$$

Here $B$ is modeled as

$$B = -\nu_T\left[\left(\frac{dU}{dz}\right)^2 + \left(\frac{dV}{dz}\right)^2\right]\frac{z}{L} = -\nu_T\mathcal{S}^2\frac{z}{L} \tag{16}$$



following MOST, using the similarity functions of Dyer (1974) as discussed in van der Laan et al. (2017). We use the flow-dependent parameter $C_{\varepsilon,3}^* \equiv 1 + \alpha_B(C_{\varepsilon,1} - C_{\varepsilon,2})$ of Sogachev et al. (2012), which for unstable conditions includes the prescription

$$\alpha_B = 1 - \left[1 + \frac{(C_{\varepsilon,2} - 1)}{(C_{\varepsilon,2} - C_{\varepsilon,1})}\right] \frac{\ell}{\ell_{\max}}, \tag{17}$$

amenable to the free-convection limit: $\varepsilon/B \to 1$ for $P/B \to 0$. Further, $\alpha_B \to 1$ as $\ell \to 0$, matching neutral conditions since $z/L$ also vanishes then. The prescription (Eq. (17)) results in

$$C_{\varepsilon,3}^* = 1 + C_{\varepsilon,1} - C_{\varepsilon,2} + (2C_{\varepsilon,2} - C_{\varepsilon,1} - 1) \frac{\ell}{\ell_{\max}}, \tag{18}$$

which also means that $C_{\varepsilon,3}^* \to C_{\varepsilon,2}$ approaching the effective ABL top ($\ell \to \ell_{\max}$), so that sources and sinks of $\varepsilon$ balance in Eq. (15); i.e. $P - \varepsilon + B$ all have the same coefficient.

## 4    Methodology of numerical simulations

The one-dimensional numerical simulations are performed with EllipSys1D (van der Laan and Sørensen, 2017a), which is a simplified one-dimensional version of EllipSys3D, initially developed by Sørensen (1994) and Mikkelsen (2003). EllipSys1D is a finite volume solver for incompressible flow, with collocated storage of flow variables. It is assumed that the vertical velocity is zero and the pressure gradients are constant, which is valid in an idealized ABL, as discussed in Section 2. As a

consequence, it is not necessary to solve the pressure correction equation that is normally used to ensure mass conservation.

### 4.1    Ambient turbulence in the limited length-scale $k$-$\varepsilon$ turbulence model

The limited length scale $k$-$\varepsilon$ model typically simulates an eddy viscosity that decays to zero for $z \to \infty$, which can lead to numerical instability. While e.g. Koblitz et al. (2015) chose to set upper limits for $k$ and $\varepsilon$ to prevent numerical instabilities, we prefer a more physical method, including ambient source terms $S_{k,\mathrm{amb}}$ and $S_{\varepsilon,\mathrm{amb}}$ to the $k$ and $\varepsilon$ transport equations,

respectively. Following Spalart and Rumsey (2007), we set

$$S_{k,\mathrm{amb}} = \varepsilon_{\mathrm{amb}}, \qquad S_{\varepsilon,\mathrm{amb}} = C_{\varepsilon,2} \frac{\varepsilon_{\mathrm{amb}}^2}{k_{\mathrm{amb}}}. \tag{19}$$

When all sources of turbulence are zero ($\mathcal{P} = B = 0$) and the diffusion terms are zero ($dk/dz = d\varepsilon/dz = 0$), then $k = k_{\mathrm{amb}}$ and $\varepsilon = \varepsilon_{\mathrm{amb}}$. To be consistent with the equations solved, we define the ambient turbulence quantities in terms of the driving parameters, $G$ and $\ell_{\max}$:

$$\ell_{\mathrm{amb}} = C_{\mathrm{amb}} \ell_{\max}, \qquad k_{\mathrm{amb}} = \frac{3}{2} I_{\mathrm{amb}}^2 G^2, \qquad \varepsilon_{\mathrm{amb}} = C_\mu^{3/4} \frac{k_{\mathrm{amb}}^{\frac{3}{2}}}{\ell_{\mathrm{amb}}} = C_\mu^{3/4} \frac{3}{2} \sqrt{\frac{3}{2}} \frac{I_{\mathrm{amb}}^3}{C_{\mathrm{amb}}} \frac{G^3}{\ell_{\max}}. \tag{20}$$

Here $I_{\mathrm{amb}}$ is the total turbulence intensity[1] above the (simulated) ABL, and $C_{\mathrm{amb}}$ is the ratio of the turbulence length scale above the ABL ($\ell_{\mathrm{amb}}$) to maximum turbulence length scale ($\ell_{\max}$). We choose small values for $I_{\mathrm{amb}} = 10^{-6}$ and $C_{\mathrm{amb}} = 10^{-6}$,

---

[1]From the two-equation $k$-$\varepsilon$ model (which is isotropic), the total turbulence intensity is calculated by $I = \sqrt{2/3k}/\sqrt{U^2 + V^2}$.



such that the ambient turbulence does not affect the solution for $U$ and $V$, while the numerical stability is maintained. It should be noted that the overshoot in $\ell/\ell_{\max}$ that can occur near the ABL depth is still affected by the ambient values. Sogachev et al. (2012) and Koblitz et al. (2015) chose to use a limiter on $\varepsilon$ to avoid an overshoot in $\ell$, but we choose not use it. In general, we prefer to avoid limiters because they can break the Rossby number similarity that is presented in Section 5.

## 4.2 Numerical setup

The flow that we are solving is relatively stiff, and we choose to include the transient terms using a second order three level implicit method with a large time step that is set as $1/|f_c|$ s. All spatial gradients are discretized by a second order central difference scheme. Convergence is typically achieved after $10^5$ iterations, which takes about 10 s on a single 2.7 GHz CPU. The domain height is set to $10^5$ m to assure that the ABL depth is significantly smaller than the domain height for all flow cases considered. The numerical grid represents a line, where the first cell height is set to $10^{-2}$ m. The cells are stretched for increasing heights using an expansion ratio of about 1.2. The grid consists of 384 cells, which is based on a grid refinement study presented in Section 4.3. A rough wall boundary conditions is set at the ground. as discussed by Sørensen et al. (2007). For the length scale limited $k$-$\varepsilon$ model, this means that we set $\varepsilon$ at the first cell, use a Neumann condition for $k$, and the shear stress at the wall is defined by the neutral surface layer. The first cell is placed on top of the roughness length, which allows us to choose the first cell height independent of the roughness length. This means that we add the roughness length to all relations that include $z$, i.e., $z + z_0$. For the limited mixing-length model, we simply set the eddy viscosity from the neutral surface layer at the first cell. Neumann conditions are set for all flow variables at the top boundary.

The turbulence model constants of the $k$-$\varepsilon$ model are set as $(C_\mu, \sigma_k, \sigma_\varepsilon, C_{\varepsilon,1}, C_{\varepsilon,1}, \kappa) = (0.03, 1.0, 1.3, 1.21, 1.92, 0.4)$. The chosen $C_\mu$ value is based on neutral ASL measurements, as discussed by Richards and Hoxey (1993), and $C_{\varepsilon,1}$ is used to maintain the neutral ASL solution of the $k$-$\varepsilon$ model.

## 4.3 Grid refinement study

A grid refinement study of the numerical setup is performed for the limited length scale $k$-$\varepsilon$ model of Apsley and Castro (1997), using 48, 96, 192, 384 and 768 cells. We choose $f_c = 10^{-4}$ s$^{-1}$, $z_0 = 10^{-4}$ m and $G = 10$ ms$^{-1}$ for $\ell_{\max} = 100$ and $\ell_{\max} = 1$ m. The results in terms of wind speed of each grid are depicted in Fig. 2 for both values of $\ell_{\max}$. For $\ell_{\max} = 100$ m, the largest difference with respect to the finest grid is 0.5 %, 0.2 %, 0.09 % and 0.03% for 48, 96, 192, 384 cells, respectively, located at the first cell near the wall boundary. When using $\ell_{\max} = 1$ m, a small ABL depth of 100 m is simulated with a sharp low level jet. In the zoomed plot of Fig. 2b, one can see how the grid size affects the low level jet, where the largest difference with respect to the finest grid is 1 %, 0.2 % and 0.04 % and 0.01 %, for 48, 96, 192, 384 cells, respectively. We find similar results for the limited mixing-length model of Blackadar (1962). In addition, the turbulence model extensions to unstable surface layer stratification typically shows smaller difference between the grids due to the enhanced mixing and the use of a high $\ell_{\max}$ value that represents a convective ABL. Hence, our choice of using 384 cells is conservative.





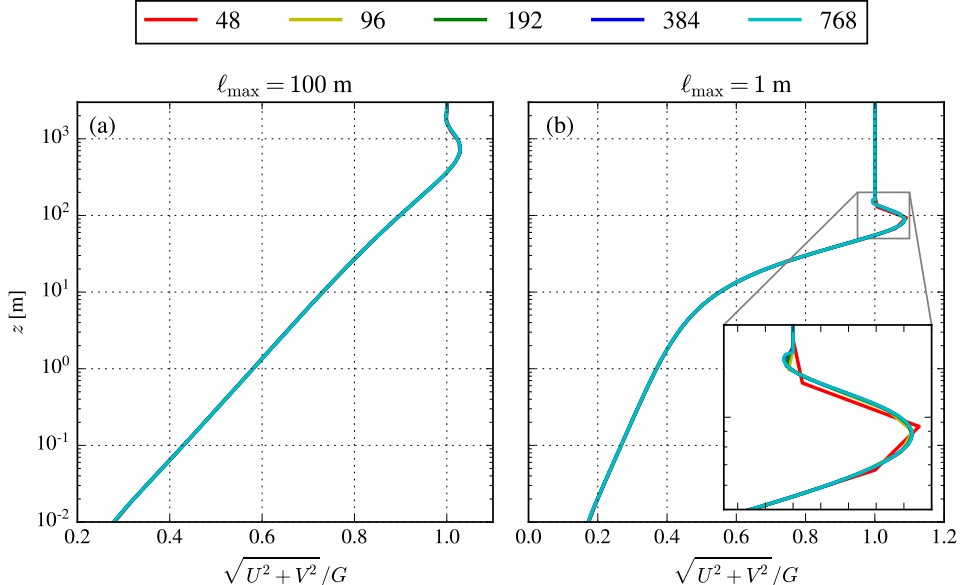

**Figure 2.** Grid refinement study of the one-dimensional RANS simulation using the limited length scale $k$-$\varepsilon$ model, for 48, 96, 192, 384 and 768 cells. **(a)** $\ell_{\max} = 100$ m. **(b)** $\ell_{\max} = 1$ m.

## 5    Rossby number similarity in numerical and analytical solutions

The numerical solution of the original limited length scale turbulence closures of Blackadar (1962) and Apsley and Castro (1997) depend on four parameters: $f_c$, $G$, $\ell_{\max}$ and $z_0$. Blackadar (1962) argued that the maximum turbulence length scale in the ABL should be proportional to the length scale $G/|f_c|$. We find that if both $\ell_{\max}$ and $z_0$ are proportional to $G/|f_c|$, then

the ABL profiles only depend on two dimensionless numbers, which can be written as two Rossby numbers with different characteristic length scales. The Rossby number, $\mathrm{Ro} = U/(|f_c|L)$, describes the ratio of the inertial force with respect to the Coriolis force, where $U$ and $L$ are characteristic velocity and length scales, respectively. We define two Rossby numbers based on the geostrophic wind $G$ as the characteristic velocity scale, with two different characteristic length scales $\ell_{\max}$ and $z_0$:

$$\mathrm{Ro}_\ell \equiv \frac{G}{|f_c|\ell_{\max}}, \qquad \mathrm{Ro}_0 \equiv \frac{G}{|f_c|z_0} \tag{21}$$

$\mathrm{Ro}_0$ is known as the surface Rossby number, first introduced by Lettau (1959); $\mathrm{Ro}_\ell$ is analogous to the reciprocal of a dimensionless boundary layer depth (e.g. Arya and Wyngaard, 1975). We define $\ell_{\max}$ and $z_0$ as:

$$\ell_{\max} = \frac{1}{\mathrm{Ro}_\ell}\frac{G}{|f_c|}, \qquad z_0 = \frac{1}{\mathrm{Ro}_0}\frac{G}{|f_c|} \tag{22}$$

Hence, we have reduced the number of dependent parameters from four to two: $f(f_c, G, \ell_{\max}, z_0) \rightarrow f(\mathrm{Ro}_\ell, \mathrm{Ro}_0)$. For a fixed surface roughness $z_0$, then the ratio of the two Rossby numbers is the only dependent parameter:

$$\ell_{\max} = \frac{\mathrm{Ro}_0}{\mathrm{Ro}_\ell}z_0; \tag{23}$$





i.e., the ratio of simulated ABL depth to $z_0$ is the lone parameter. Blackadar (1962) found a characteristic maximum ABL turbulence length scale of $0.00027G/|f_c|$ for the Leipzig wind profile (Lettau, 1950), which equating with $\ell_{\max}$ corresponds to $\mathrm{Ro}_\ell \simeq 3700$.

Figure 3 depicts the Rossby number similarity of our one-dimensional RANS simulations using the original limited length

scale turbulence closures of Blackadar (1962), Fig. 3a-c, and Apsley and Castro (1997), Fig. 3d-g. Four combinations of $\mathrm{Ro}_0$ ($10^6$ and $10^9$) and $\mathrm{Ro}_\ell$ ($10^3$ and $10^5$) are used, each simulated with four combinations of $G$ (10 and 20 ms$^{-1}$) and $f_c$ ($5 \times 10^{-5}$ and $10^{-4}$ s$^{-1}$). The roughness length and maximum turbulence length scale follow from Eq. (22) and cover a wide range from $z_0 = 10^{-4} - 0.4$ m and from $\ell_{\max} = 1 - 4 \times 10^2$ m. Figure 3 shows that normalized wind speed, wind direction and turbulence quantities for both turbulence closures are only dependent of $\mathrm{Ro}_0$ and $\mathrm{Ro}_\ell$. Both turbulence closures produces similar results in

terms of wind speed, wind direction and eddy-viscosity. The limited length scale $k$-$\varepsilon$ model of Apsley and Castro (1997) also predicts a total turbulence intensity $I$ (Fig. 3g) and a turbulence length scale $\ell$ (not shown in Fig. 3), which are only dependent on the two Rossby numbers. In addition, the total turbulence intensity close to the surface only depends on $\mathrm{Ro}_0$, while further away, it is mainly influenced by $\mathrm{Ro}_\ell$ with a weaker dependence on $\mathrm{Ro}_0$.

Considering the non-neutral ABL with Coriolis effects but ignoring the strength of capping-inversion (entrainment), in

the micrometeorological literature the Kazanski-Monin (1961) parameter $u_{*0}/(|f_c|L)$ is typically invoked (e.g. Arya, 1975; Zilitinkevich, 1989). This can also be considered like a third Rossby number, which in our context of using $G$ instead of $u_{*0}$ is

$$\mathrm{Ro}_{L_-} \equiv \frac{-G}{|f_c|L}; \tag{24}$$

here the subscript ($_{L_-}$) denotes that Eq. (24) is defined for unstable conditions, i.e. $L \leq 0$. For the convective boundary layer, $u_{*0}/(-|f_c|L)$ is generally replaced by the dimensionless inversion height $-z_i/L$, because the convective ABL depth does not

have a significant dependence on $u_{*0}/f_c$ (Arya, 1975). However, we note that $\mathrm{Ro}_{L_-}$ functions as a 'bottom-up' parameter in the non-neutral RANS equation set, with the Obukhov length $L$ in Eq. (16) specified as a surface-layer quantity; in effect $\mathrm{Ro}_{L_-}$ dictates the relative increase in mixing-length (i.e. in the dimensionless coordinate $z|f_c|/G$). Our length scale limited turbulence closures extended to unstable surface layer stratification, as presented in Section 3, are dependent on $\mathrm{Ro}_{L_-}$. For $\mathrm{Ro}_{L_-} = 0$, the extended models return to the original models. Figure 4 depicts the Rossby number similarity of the extended

turbulence closures using six combinations of the three Rossby numbers, which are each simulated with four combinations of $G$ and $f_c$. We use two values of $\mathrm{Ro}_0$ ($10^6$ and $10^9$) and three values of $\mathrm{Ro}_{L_-}$ ($0$, $5 \times 10^2$ and $2 \times 10^3$) for $\mathrm{Ro}_\ell = 10^3$. For these Rossby number combinations, $\mathrm{Ro}_{L_-} = 5 \times 10^2$ and $\mathrm{Ro}_{L_-} = 2 \times 10^3$ correspond to near-unstable conditions ($-1/L = 0.00125$-$0.005$ m$^{-1}$) and unstable to very unstable conditions ($-1/L = 0.005$–$0.02$ m$^{-1}$), respectively. Figure 4 shows the both extended turbulence closures only depend on $\mathrm{Ro}_0$, $\mathrm{Ro}_{L_-}$, for a given $\mathrm{Ro}_\ell$. Although not shown in Fig. 4, changing $\mathrm{Ro}_\ell$ would not

break the Rossby number similarity. Note that it does not make sense to include combinations of non zero values of $\mathrm{Ro}_{L_-}$ that correspond to unstable conditions and large values of $\mathrm{Ro}_\ell$ that corresponds to stable conditions.

The extended limited length scale mixing-length model (Fig. 4a-c) is less sensitive to $\mathrm{Ro}_{L_-}$ compared to the extended limited length scale $k$-$\varepsilon$ model (Fig. 4d-g) because of the buoyancy production in the transport equations of $k$ and $\varepsilon$, which is not present in the extended mixing-length model. Both models predict a deeper ABL (larger $z_i$) that is more mixed, for stronger





**Figure 3.** Rossby number similarity of the original turbulence closures. **(a-c)** Limited mixing-length model. **(d-g)** Limited length scale $k$-$\varepsilon$ model.

unstable surface layer stratification (increasing $\mathrm{Ro}_{L_-}$). The wind veer is also reduced for stronger unstable conditions for the extended $k$-$\varepsilon$ model (Fig. 4e), but it does not always decrease for increasing unstable conditions for the extended mixing-length model (Fig. 4b).



| $\mathrm{Ro}_0, \mathrm{Ro}_\ell, \mathrm{Ro}_{L_-}$ | $f_c[\mathrm{s}^{-1}], G[\mathrm{ms}^{-1}]$ | $f_c[\mathrm{s}^{-1}], G[\mathrm{ms}^{-1}]$ | $f_c[\mathrm{s}^{-1}], G[\mathrm{ms}^{-1}]$ | $f_c[\mathrm{s}^{-1}], G[\mathrm{ms}^{-1}]$ |
|---|---|---|---|---|
| $10^6, 10^3, 0$: | $5\times10^{-5}, 10$ | $5\times10^{-5}, 20$ | $10^{-4}, 10$ | $10^{-4}, 20$ |
| $10^9, 10^3, 0$: | $5\times10^{-5}, 10$ | $5\times10^{-5}, 20$ | $10^{-4}, 10$ | $10^{-4}, 20$ |
| $10^6, 10^3, 5\times10^2$: | $5\times10^{-5}, 10$ | $5\times10^{-5}, 20$ | $10^{-4}, 10$ | $10^{-4}, 20$ |
| $10^6, 10^3, 2\times10^3$: | $5\times10^{-5}, 10$ | $5\times10^{-5}, 20$ | $10^{-4}, 10$ | $10^{-4}, 20$ |
| $10^9, 10^3, 5\times10^2$: | $5\times10^{-5}, 10$ | $5\times10^{-5}, 20$ | $10^{-4}, 10$ | $10^{-4}, 20$ |
| $10^9, 10^3, 2\times10^3$: | $5\times10^{-5}, 10$ | $5\times10^{-5}, 20$ | $10^{-4}, 10$ | $10^{-4}, 20$ |

**Figure 4.** Rossby number similarity of the turbulence closures extended to unstable surface layer conditions. **(a-c)** Limited mixing-length model. **(d-g)** Limited length scale $k$-$\varepsilon$ model.





One could choose to use the friction velocity at the surface, $u_{*0}$, as a velocity scale in the Rossby numbers instead of the geostrophic wind speed. However, the friction velocity depends on height $z$, and is a result of the model, not an input. In other words, the height at which the friction velocity needs to be extracted to get a collapse is also dependent on the ABL profiles, since the height scales with friction velocity. Hence it is more sensible to use geostrophic wind speed as a velocity scale in the

model-based Rossby number similarity—consistent also with classic Ekman theory (which relates the wind speed in terms of $G$).

The Rossby number similarity can be employed to generate a library of ABL profiles for a range of $\mathrm{Ro}_0$, $\mathrm{Ro}_\ell$ and $\mathrm{Ro}_{L_-}$. The library contains all possible model solutions for the range of chosen Rossby numbers and it can be used to determine inflow profiles for three-dimensional RANS simulations, without the need of running one-dimensional precursor simulations.

The obtained Rossby number similarity can only be achieved for a grid independent numerical setup, as we have shown in Section 4.3. In addition, the ambient source terms should also be scaled by the relevant input parameters ($G$ and $\ell_{\mathrm{max}}$), as discussed in Section 4.1.

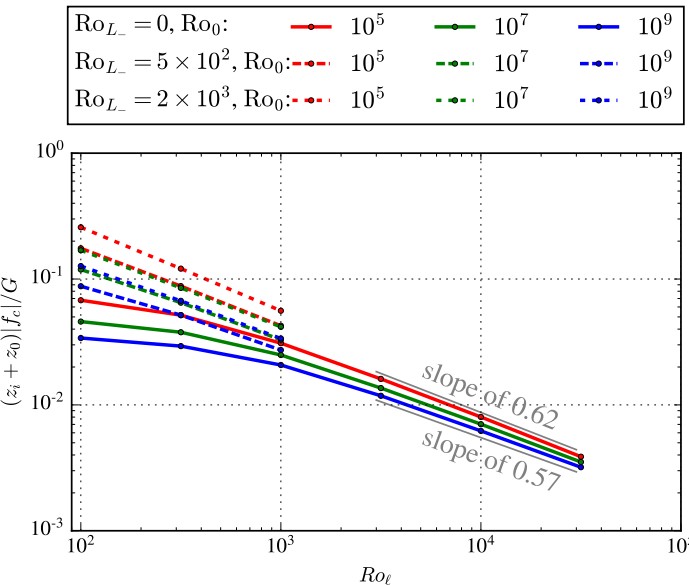

**Figure 5.** Normalized boundary layer depth $z_i$ predicted by limited length scale $k$-$\varepsilon$ model extended to unstable surface layer stratification, as function of the three Rossby numbers.

The ABL depth $z_i$ predicted by the original limited length scale turbulence closures are mainly dependent on the maximum turbulence length scale $\ell_{\mathrm{max}}$. The normalized ABL depth ($(z_i + z_0)|f_c|/G$) is mainly dependent on $\mathrm{Ro}_\ell$, which is depicted

in Fig. 5, where results of the limited length scale $k$-$\varepsilon$ model extended to unstable surface layer stratification are shown for $3 \times 6 \times 3$ combinations of the three Rossby numbers $\mathrm{Ro}_0$, $\mathrm{Ro}_\ell$ and $\mathrm{Ro}_{L_-}$. We have chosen $G = 10$ ms$^{-1}$ and $f_c = 10^{-4}$ s$^{-1}$, but the results are independent of $G$ and $f_c$ due to the Rossby number similarity. The normalized ABL depth is defined as



the height at which the wind direction (relative to the ground) becomes zero for the second time, i.e. above the mean jet and associated turning as in Ekman theory. For the Ekman solution (Section A1), this definition results in an ABL depth equal to $z_i = 2\pi\sqrt{2\nu_T/|f_c|}$. The normalized ABL depth in the RANS model increases for stronger unstable surface layer conditions (larger $\mathrm{Ro}_{L_-}$), i.e. for larger values of the surface heat flux. For neutral and stable conditions ($\mathrm{Ro}_{L_-} = 0$) and

moderate to shallow ABL depths, i.e. $3 \times 10^3 \leq \mathrm{Ro}_\ell \leq 3 \times 10^4$—corresponding to $z_i <\sim 2000\,\mathrm{m}$ as seen in Fig. 5—we find that $\log_{10}([z_i + z_0]|f_c|/G) \propto a\log_{10}(\mathrm{Ro}_\ell)$, with $a = 0.57$–$0.62$ for $\mathrm{Ro}_0$ over the range of $10^9$–$10^5$. Hence for moderate to shallow ABLs the effective depth modeled in neutral and stable conditions is roughly $z_i \propto \ell_{\max}^a (G/|f_c|)^{1-a}$, with $a \approx 0.6$. As seen by the solid lines in Fig. 5, under neutral conditions and large ABL depths, the $z_i$ dependence on $\ell_{\max}$ softens ($a < 2/3$) and deviates from a power law, while for unstable conditions $a$ is similar to the previously found value of $0.6$.

## 10  6  Validation and model limits

We employ the Rossby similarity from Section 5 to validate a range of results simulated by the original limited length scale $k$-$\varepsilon$ model of Apsley and Castro (1997) including our proposed extension to unstable surface layer stratification. Historical measurements of the geostrophic drag coefficient $u_{*0}/G$ and the cross isobar angle (the angle between the surface wind direction and the geostrophic wind direction), as summarized by Hess and Garratt (2002), and measured profiles of the ASL

and ABL for different atmospheric stabilities from Peña et al. (2010) and Peña et al. (2014), respectively, are used as validation metrics. The limited mixing-length model of Blackadar (1962) and its extension are not considered in the comparison with measurements, since we are mainly interested in the $k$-$\varepsilon$ model.

### 6.1  Geostrophic drag coefficient

The geostrophic drag law (GDL) is a widely used relation in boundary-layer meteorology and wind resource assessment (after

Troen and Petersen, 1989), which connects the surface layer properties as $z_0$ and $u_{*0}$ with the driving forces on top of the ABL proportional to $|f_c|G$:

$$G = \frac{u_{*0}}{\kappa}\sqrt{\left[\ln\left(\frac{u_{*0}}{|f_c|z_0}\right) - A\right]^2 + B^2},\tag{25}$$

where $A$ and $B$ are empirical constants. The GDL can be derived from Eq. (1), where the Reynolds-stresses do not need to be modelled explicitly (as in e.g. Zilitinkevich, 1989), and can be expressed as an implicit relation for the geostrophic drag

coefficient $u_{*0}/G$ and $\mathrm{Ro}_0$:

$$\frac{u_{*0}}{G} = \frac{\kappa}{\sqrt{\left[\ln\left(\mathrm{Ro}_0\right) + \ln\left(\frac{u_{*0}}{G}\right) - A\right]^2 + B^2}}.\tag{26}$$

Figure 6 is a reproduction from Hess and Garratt (2002), where the geostrophic drag coefficient is depicted as function of surface Rossby number $\mathrm{Ro}_0$. The black markers are measurements summarized by Hess and Garratt (2002), where the dots are near-neutral and near-barotropic conditions, the triangles and squares reflect less idealized atmospheric conditions and the open





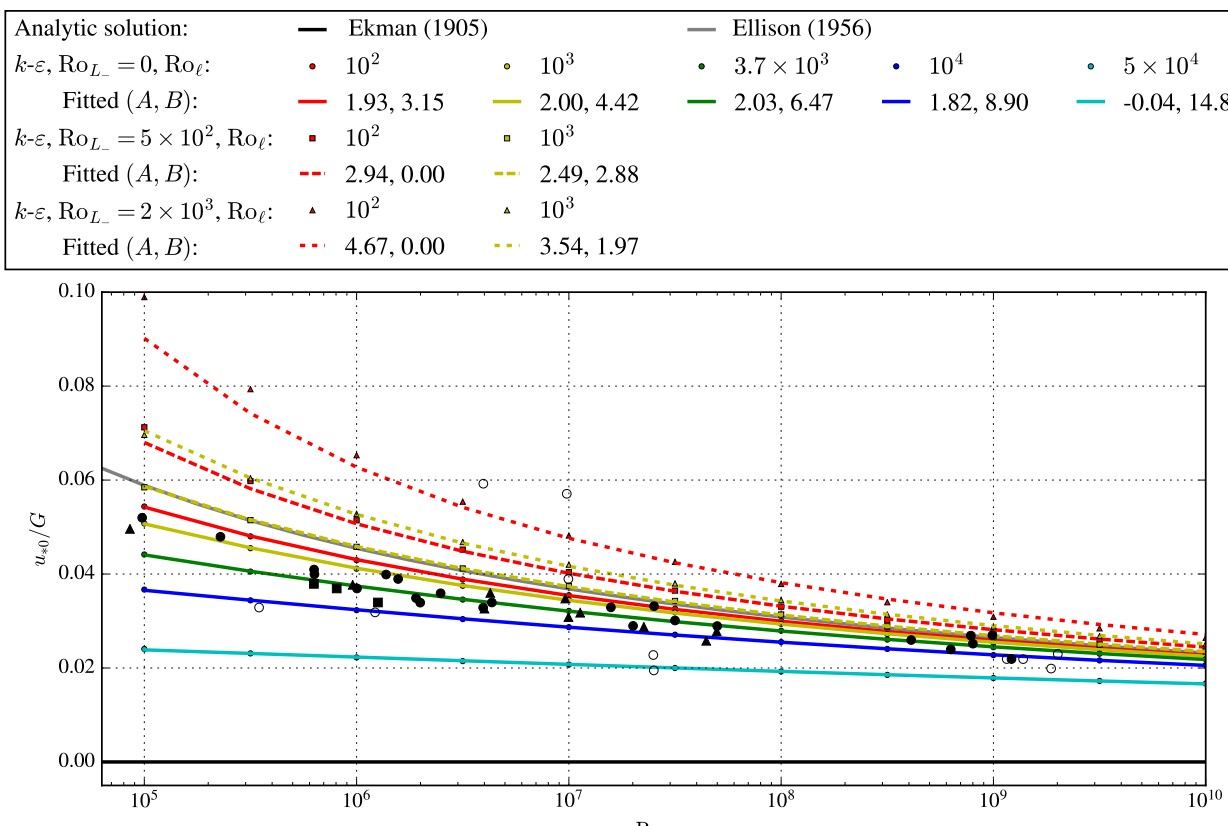

**Figure 6.** Reproduced from Hess and Garratt (2002). Geostrophic drag coefficient simulated by the limited length scale $k$-$\varepsilon$ model extended to unstable surface layer stratification, taken at a normalized height of $(z+z_0)|f_c|/G = 5\times10^{-5}$ (i.e., in the surface layer), for different $\mathrm{Ro}_0$, $\mathrm{Ro}_\ell$ and $\mathrm{Ro}_{L_-}$. Black markers represent measurements from Hess and Garratt (2002). $\mathrm{Ro}_\ell = 3.7 \times 10^3$ represents $\ell_{\max}$ from Blackadar (1962). Analytic results of Ekman (1905) and Ellison (1956) are summarized in Appendix A

.

circles are measurements with a relative high uncertainty. Results of the limited length scale $k$-$\varepsilon$ model including the extension to unstable surface layer stratification are shown as colored markers, where the colors represent a range of $\mathrm{Ro}_\ell$. For the two smallest values of $\mathrm{Ro}_\ell$, two additional results are plotted for $\mathrm{Ro}_{L_-} = 5 \times 10^2$ and $\mathrm{Ro}_{L_-} = 2 \times 10^3$, representing unstable $(-1/L = 0.005\ \mathrm{m}^{-1})$ and very unstable conditions $(-1/L = 0.02\ \mathrm{m}^{-1})$ for the chosen values of $G = 10\ \mathrm{ms}^{-1}$ and $f_c = 10^{-4}\ \mathrm{s}^{-1}$.

5 The colored lines are fitted $A$ and $B$ constants from the GDL as defined in Eq. (26). The analytic solutions from Ekman (1905) and Ellison (1956), as summarized in Appendix A, are shown as black and gray lines, respectively. For $\mathrm{Ro}_{L_-} = 0$, the geostrophic drag coefficient predicted by the limited length scale $k$-$\varepsilon$ model is bounded by the analytic solutions. For $\mathrm{Ro}_\ell \to 0$, the geostrophic drag coefficient of Ellison (1956) is approximated. For increasing $\mathrm{Ro}_\ell$ or decreasing ABL depths, the $\{u_{*0}/G, \log(\mathrm{Ro}_0)\}$ relationship becomes more linear. In addition, for $\mathrm{Ro}_\ell = 3.7 \times 10^3$, as used by Blackadar (1962), and



$\mathrm{Ro}_{L_-} = 0$, most of the near-neutral and near-barotropic measurements are captured quite well. Hess and Garratt (2002) used the measurements of the geostrophic drag coefficient to validate a number of models, which often have only one result for each $\mathrm{Ro}_0$. The geostrophic drag coefficients predicted by the limited length scale $k$-$\varepsilon$ model can cover all measurements by varying $\mathrm{Ro}_\ell$. In addition, the extension to unstable surface layer conditions, can also explain the trend of the more uncertain

measurements (black dots). Since $\mathrm{Ro}_\ell$ and $\mathrm{Ro}_{L_-}$ influence the ABL depth, as previously shown in Fig. 5, the model suggests that the measurements were conducted for a range of ABL depths that could reflect a range of atmospheric stabilities, although the geostrophic wind shear can play a role here as shown by Floors et al. (2015).

The fitted $A$ and $B$ parameters in Fig. 6 are dependent on $\mathrm{Ro}_\ell$ and $\mathrm{Ro}_{L_-}$, which both influence the ABL depth. Typically used values in wind energy are $A = 1.8$ and $B = 4.5$ (e.g. Troen and Petersen, 1989), which quite closely matches the blue line

($\mathrm{Ro}_\ell = 10^4, A = 1.82, B = 8.90$). Assuming $\ell_{\max}$ is a measure of the ABL depth, then in the actual atmosphere over land we have $\mathrm{Ro}_0/\mathrm{Ro}_\ell \sim 10^3$–$10^5$, while over sea the ratio is roughly $10^6$–$10^7$. Thus one can see that the typical wind energy values of $A$ and $B$ are a compromise for applicability over both land and sea. The real-world limits mean that the result for $\mathrm{Ro}_\ell = 10^2$ (red line) can extend only from $\mathrm{Ro}_0 \sim 10^5$–$10^7$, while the oversea regime (large $\mathrm{Ro}_0$) tends to involve a smaller range of $\mathrm{Ro}_\ell$. We remind that the GDL from Eq. (25) limits how large $B$ can be; generally $u_{*0}/G < \kappa/B$, so values of $B$ greater than those

shown are not physical. The model results in Fig. 6 do not violate this limit.

## 6.2 Cross isobar angle

Figure 7 is a reproduction of Hess and Garratt (2002), where the angle between surface wind direction and the geostrophic wind direction is plotted as function of the surface Rossby number. This angle is known as the cross isobar angle, $\theta_0$. The black markers, analytic solutions and model results follow the same definition as used in Fig. 6, where additional black diamond

markers are added that correspond to climatological measurements, as discussed by Hess and Garratt (2002). For $\mathrm{Ro}_{L_-} = 0$, the model results of the cross isobar angle are bounded by the analytic solutions, as also found for the geostrophic drag coefficient in Fig. 6. All measurements summarized by Hess and Garratt (2002) can be simulated by the limited length scale $k$-$\varepsilon$ model by varying the ABL depth using $\mathrm{Ro}_\ell$. Most of the measurements are well predicted for $\mathrm{Ro}_{L_-} = 0$ and $\mathrm{Ro}_\ell = 10^3$–$10^4$, which is the range used by Blackadar (1962) ($\mathrm{Ro}_\ell = 3.7 \times 10^3$). For $\mathrm{Ro}_{L_-} \neq 0$, smaller values of the cross isobar angle can

be simulated compared the analytic solution of Ellison (1956) due to the enhanced rate of mixing. The model cannot predict larger values of the cross isobar angle compared to the analytic solution of Ekman (1905) ($45°$).

## 6.3 Atmospheric surface layer profiles

Peña et al. (2014) used measurements of the wind speed components from 10 to 160 m, from The National Test Station for Wind Turbines at Høvsøre, a coastal site in Denmark, characterized as flat grassland. The Coriolis parameter for the test

location is $1.21 \times 10^{-4}$ s$^{-1}$. The measurements where taken from sonic anemometers over one year, and a wind direction sector was selected to avoid the influence of the coastline and wind turbine wakes. Peña et al. (2014) also calculated a 'mixing'



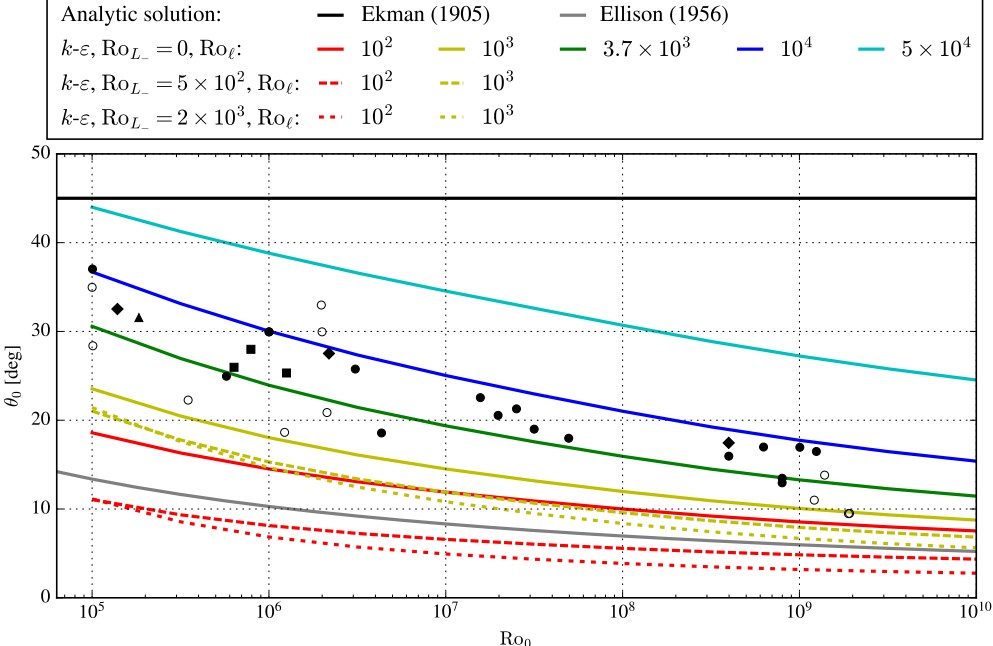

**Figure 7.** Reproduced from Hess and Garratt (2002). Cross isobar angle simulated by the limited length scale $k$-$\varepsilon$ model extended to unstable surface layer stratification, taken at a normalized height of $(z + z_0)|f_c|/G = 5 \times 10^{-5}$ for different $\mathrm{Ro}_0$, $\mathrm{Ro}_\ell$ and $\mathrm{Ro}_{L_-}$. Black markers represent measurements from Hess and Garratt (2002). $\mathrm{Ro}_\ell = 3.7$ represents $\ell_{\max}$ from Blackadar (1962). Analytic results of Ekman (1905) and Ellison (1956) are summarized in Appendix A.

(turbulence) length scale $\hat{\ell}$ using a local friction velocity $u_*$ and the wind speed gradient:

$$\hat{\ell} = \frac{u_*}{dU/dz}. \tag{27}$$

Seven cases were defined based on the atmospheric stability, and these are listed in Table 2 in terms of the Obukhov length, roughness length and friction velocity. In order to apply the limited length scale $k$-$\varepsilon$, we need to set the geostrophic wind

5  speed and the maximum turbulence length scale, which are both unknown. We choose to use $G$ and $\ell_{\max}$ as free parameters, which we fit for a reference wind speed and a turbulence length scale, at a reference height of 60 m. The wind speed gradient is obtained from a central difference scheme taking the wind speed at 40, 60 and 80 m. The fitted parameters are obtained by running the numerical simulations with a gradients based optimizer, and the results are listed in Table 2. The maximum $\ell_{\max}$ is set to $10^3$ m, which corresponds to an ABL depth on the order of 5 km, as depicted in Fig. 5. The unstable cases are also

10  simulated with the extended limited length scale $k$-$\varepsilon$ model using the measured $L$, and re-fitted $G$ and $\ell_{\max}$, which are listed in Table 2 as values in parenthesis.

Fig. 8 depicts the wind speed and turbulence length scale of the measurements and numerical simulations using the original and extended limited length scale $k$-$\varepsilon$ models. The turbulence length scale from the numerical simulation is calculated by Eq. (27), instead of the usual definition $\ell = C_\mu^{3/4} k^{3/2}/\varepsilon$. The original limited length scale $k$-$\varepsilon$ model of Apsley and Castro



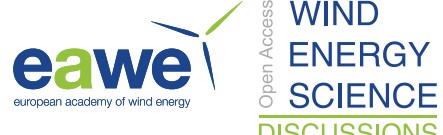

| | Data | | | Model | | |
|---|---|---|---|---|---|---|
| | $1/L$ | $z_0$ | $u_{*0}$ | Fitted $G$ | Fitted $\ell_{\max}$ | $u_{*0}$ |
| Case | [m⁻¹] | [m] | [ms⁻¹] | [ms⁻¹] | [m] | [ms⁻¹] |
| Very unstable (vu) | $-1.35 \times 10^{-2}$ | $1.3 \times 10^{-2}$ | 0.35 | 8.00 (7.50) | $10^3$ $(5.39 \times 10^2)$ | 0.30 (0.34) |
| Unstable (u) | $-7.04 \times 10^{-3}$ | $1.2 \times 10^{-2}$ | 0.41 | 10.1 (9.56) | $10^3$ $(5.54 \times 10^2)$ | 0.37 (0.40) |
| Near unstable (nu) | $-3.18 \times 10^{-3}$ | $1.2 \times 10^{-2}$ | 0.40 | 10.3 (10.0) | $10^3$ $(2.00 \times 10^2)$ | 0.37 (0.39) |
| Neutral (n) | $1.87 \times 10^{-4}$ | $1.3 \times 10^{-2}$ | 0.39 | 11.0 | $4.01 \times 10^1$ | 0.37 |
| Near stable (ns) | $3.14 \times 10^{-3}$ | $1.2 \times 10^{-2}$ | 0.36 | 11.3 | $1.72 \times 10^1$ | 0.35 |
| Stable (s) | $9.61 \times 10^{-3}$ | $0.8 \times 10^{-2}$ | 0.26 | 9.96 | $6.49 \times 10^0$ | 0.27 |
| Very stable (vs) | $3.57 \times 10^{-2}$ | $0.2 \times 10^{-2}$ | 0.16 | 8.62 | $3.35 \times 10^0$ | 0.20 |

**Table 2.** ASL validation cases. Fitted $G$, fitted $\ell_{\max}$ and modeled $u_{*0}$ in parenthesis represent values for extended model.

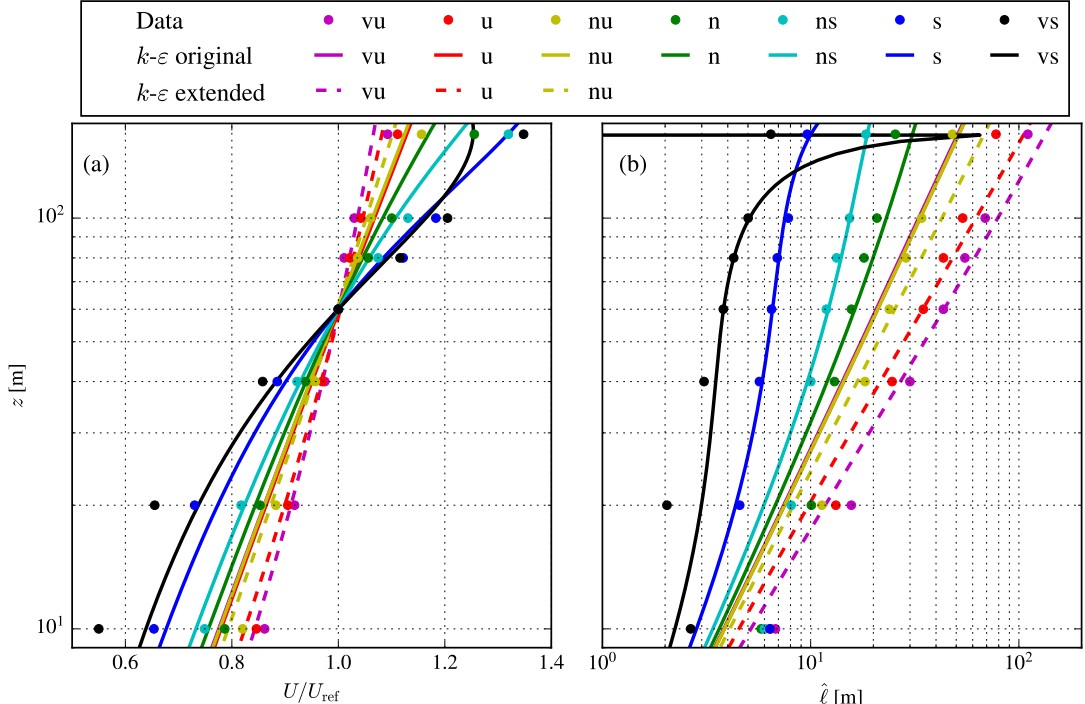

**Figure 8.** ASL measurements of Peña et al. (2010) compared to simulation results of the original limited length scale $k$-$\varepsilon$ model of Apsley and Castro (1997). **(a)** Wind speed. **(b)** Turbulence length scale from Eq. (27). Unstable cases are also simulated with our extension to unstable surface layer stratification with $L$ from Table 2.

(1997) can capture the wind speed and turbulence length scale for the stable and neutral cases. Note that for the very stable case, the shear is under estimated by the model. As expected, the original limited length scale $k$-$\varepsilon$ model cannot predict a lower





shear and a larger turbulence length scale compared to neutral atmospheric conditions (where $dU/dz = u_*/\ell$ and $\ell = \kappa z$), and the optimizer used to fit $G$ and $\ell_{\max}$ sets $\ell_{\max}$ to our chosen maximum value of $10^3$ m. Note that therefore the lines corresponding to unstable conditions of the original $k$-$\varepsilon$ model largely overlap in Fig. 8. Higher values of $\ell_{\max}$ would not improve the results. The limited length scale $k$-$\varepsilon$ model extended to unstable surface layer stratification is able to predict

turbulence length scales larger than $\ell = \kappa z$, and shows improved results for both the shear and the turbulence length scale.

Table 2 also shows the measured and simulated friction velocity at a height of 10 m. The simulated friction velocity is calculated as $u_* = (\overline{u'w'}^2 + \overline{v'w'}^2)^{1/4}$. For the unstable cases, it clear that the extended model predicts friction velocities that are closer to the measurements compared to the original limited length scale $k$-$\varepsilon$ model due to the enhanced mixing.

It should be noted that the validation presented in Fig. 8 could be considered as best possible simulation-to-measurement

comparison because we have allowed ourselves to tune both $G$ and $\ell_{\max}$. When $G$ is provided by the measurements, it is more difficult to obtain a good match, as shown in Section 6.4.

### 6.4    Atmospheric boundary layer profiles

Peña et al. (2014) performed lidar measurements of the horizontal wind speed components from 10 to 1200 m at the same test site as discussed in Section 6.3. Ten cases were selected by Peña et al. (2014) that differ in geostrophic forcing and atmospheric

stability. The cases were selected to challenge the validation of numerical models. Since our numerical setup can only handle a constant geostrophic wind speed, we select the barotropic cases from Peña et al. (2014): Cases 4, 5 and 9, and the corresponding values of the Obukhov length, geostrophic wind, roughness length, friction velocity, $\mathrm{Ro}_0$ and $\mathrm{Ro}_{L_-}$ are listed in Table 3. For convenience, we keep the case names as introduced by Peña et al. (2014). Cases 4 and 5 represent a stable and a neutral ABL with high forcing, respectively, where $\mathrm{Ro}_0 = 10^7$. Case 9 is characterized by a low forcing and very unstable stratification, where $\mathrm{Ro}_0 = 2.8 \times 10^6$.

| Case | Description | $1/L$ | $G$ | $z_0$ | $u_{*0}$ | $\mathrm{Ro}_0$ | $\mathrm{Ro}_{L_-}$ |
|------|-------------|-------|-----|-------|----------|------------------|----------------------|
|      |             | [m$^{-1}$] | [ms$^{-1}$] | [m] | [ms$^{-1}$] | [-] | [-] |
| 4 | Stable, strongly forced | $4.5 \times 10^{-3}$ | 20.5 | $1.6 \times 10^{-2}$ | 0.45 | $1.0 \times 10^7$ | - |
| 5 | Neutral | $-5 \times 10^{-4}$ | 19.5 | $1.6 \times 10^{-2}$ | 0.70 | $1.0 \times 10^7$ | - |
| 9 | Very unstable, weak forcing | $-4.0 \times 10^{-2}$ | 5.02 | $1.6 \times 10^{-2}$ | 0.26 | $2.8 \times 10^6$ | $1.7 \times 10^3$ |

**Table 3.** ABL validation cases based on Peña et al. (2014).

In Case 6 from Peña et al. (2014) it is observed that the lidar measurements do not approach the geostrophic wind speed at large heights above the surface. This is because the geostrophic wind speed in Peña et al. (2014) is derived from outputs of the Weather Research and Forecasting (WRF) model over a large area, potentially leading to a bias. Therefore, we use a slightly different approach to estimate the geostrophic wind; because the wind speed above the ABL is nearly always in geostrophic

balance we can just assume the wind speed measured by the wind lidar above the boundary layer depth to be equal to the geostrophic wind speed, thereby avoiding possible prediction errors in wind speed from the WRF model. Instead, only the





ABL depth is estimated from the WRF model outputs. The ABL depth is available as a diagnostic variable predicted by the YSU ABL scheme (Hong et al., 2006) in WRF. To be sure that the lidar wind speed is close to the geostrophic wind speed, we always estimate it from the level that is higher than the modelled ABL depth during all 30-min means, which constitute the three cases.

**Figure 9.** ABL measurements from Peña et al. (2014) compared to simulation results of the original limited length scale $k$-$\varepsilon$ model of Apsley and Castro (1997), for a range of $\mathrm{Ro}_\ell$. **(a, c)** Wind speed. **(b, d)** Wind direction. Unstable Case 9 is also simulated with our extension to unstable surface layer stratification with $\mathrm{Ro}_{L_-}$ (or $L$) from Table 3.



Since $G$ is known, we can use the Rossby similarity for the model validation. We could try to find an $\ell_{\max}$ to get the best comparison with the measurements, but we find that it is difficult to define a good metric. For example, we could try to find an $\ell_{\max}$ that results in the ABL depth from the measurement cases; however, the ABL depth was not directly measured and only estimated from a model. Instead of finding a single $\ell_{\max}$ value, we choose to simulate a range of $\ell_{\max}$ values.

Figure 9 depicts the measured wind speed and wind direction, for each validation case. Since Cases 4 and 5 have the same $G$ (within 5%) and thus same surface Rossby number $\mathrm{Ro}_0 \simeq 10^7$, we can plot them together because the normalized model results are the same for both cases. The error bars represent the standard error of the mean. The original limited length scale $k$-$\varepsilon$ model of Apsley and Castro (1997) is employed with a range of $\mathrm{Ro}_\ell$. The unstable ABL case (Case 9) is also simulated with the model extension to unstable surface layer stratification using $\mathrm{Ro}_{L_-}$ from Table 3 and the two smallest values of $\mathrm{Ro}_\ell$. Case

4 has a strong wind shear and a wind veer that leads to a cross isobar angle of $50°$. The limited length scale $k$-$\varepsilon$ model can predict a maximum cross isobar angle of $45°$ for extremely shallow ABL depths, as shown in Section 6.2. Hence, the measured ABL from Case 4 is not a possible numerical solution. The measured ABL from Case 5 can be predicted by the original limited length scale $k$-$\varepsilon$ model, while this is not the case for the wind speed close to the ground of Case 9 due to the strong unstable stratification. When the limited length scale $k$-$\varepsilon$ model including the extension for unstable surface layer conditions

is employed, the prediction of the wind speed near the ground is improved, although it is difficult to both get a correct wind speed and wind direction. It should be noted that the extended model only improves the wind speed near ground at $10\,\mathrm{m}$. From the measurements during Case 9 it was observed that the WRF-modeled ABL depth grew from $300\,\mathrm{m}$ to nearly $1200\,\mathrm{m}$, which indicates that the conditions were largely transient; such non-stationary conditions are difficult for a RANS model. More unstable cases are necessary to further validate the extended model, including measurements of turbulence quantities such as

the (total) turbulence intensity. It is possible to use validation cases based on turbulence-resolving methods, such as large eddy simulation, in future work.

## 7  Conclusions

The idealized ABL was simulated with a one-dimensional RANS solver, using two different turbulence closures: a limited mixing-length model, and a limited length scale $k$-$\varepsilon$ model. While these models require four input parameters, we have shown

that the simulated ABL profiles collapse to a dependence upon two Rossby numbers, which correspond to the roughness length and the maximum turbulence length scale, respectively. The Rossby number based on the maximum turbulence length scale is a new dimensionless number and is related to the ABL depth. The model-based Rossby number similarity obtained herein is valid for both turbulence models. We have employed the Rossby number similarity to compare the range of model solutions with historical measurements of relevant associated meteorological quantities, such as the geostrophic drag coefficient and

cross-isobar angle. The measured variation in these measurements can be explained by dependence upon the new Rossby number. In addition, we have shown how two classic analytic solutions of the idealized ABL (Ekman, 1905; Ellison, 1956) act as bounds on the results obtainable by the limited length scale $k$-$\varepsilon$ model.



The limited length scale turbulence closures can represent the effects of stable and neutral stratification, but cannot model unstable conditions. We have proposed simple extensions to overcome this issue, without adding a temperature equation (van der Laan et al., 2017). The extended models require an additional input, the Obukhov length, which can also be written as a third Rossby number. We have shown that the extension of the $k$-$\varepsilon$ model compares well with measurements of seven ASL profiles, representing a range of atmospheric stabilities, including three unstable cases. The $k$-$\varepsilon$ model further offers turbulence intensity, whose profile is also found to collapse according to the developed similarity theories. A model validation of the full ABL for a stable, a neutral and an unstable case is performed, with less success for the non-neutral cases. More validation cases based on the convective ABL are necessary to quantify the performance of the turbulence model extension to unstable conditions beyond the surface layer.

The application of the one-dimensional RANS simulations to generate inflow profiles for three-dimensional RANS simulations are not performed here and it should be investigated in future work.

*Code and data availability.* The numerical results are generated with proprietary software, although the data presented can be made available by contacting the corresponding author.

*Author contributions.* MPVDL has performed the simulations, obtained the model-based Rossby number similarity for the $k$-$\varepsilon$ model, produced all figures and drafted the article. MPVDL and MK proposed the extension to unstable conditions. MK added connections and relations to meteorological theory, and interpretations. AP proposed the Rossby number similarity of the mixing-length model. AP and RF contributed to the model validation. All authors contributed to the methodology and finalization of the paper.

*Competing interests.* The authors declare that they have no conflict of interest.





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

## Appendix A: Analytic solutions of the idealized ABL

### A1 Constant eddy viscosity: Ekman spiral

The analytic solution of Ekman (1905), known as the Ekman spiral, can be expressed as a function of a single variable, the normalized height $\xi \equiv z\sqrt{|f_c|/(2\nu_T)}$. The wind speed $S = \sqrt{U^2 + V^2}$ and the wind direction $\theta$ can be written as:

$$S(\xi) = G\sqrt{1 - 2\cos(\xi)\exp(-\xi) + \exp(-2\xi)}, \qquad \theta(\xi) = \arctan\left(\frac{\sin(\xi)}{\exp(\xi) - \cos(\xi)}\right) \tag{A1}$$

The cross isobar angle, $\theta_0$, is found to be $45°$ and the geostrophic drag coefficient is zero.





## A2 Linear eddy viscosity: Ellison

The analytic solution of Ellison (1956) for the $U$ and $V$ velocity components can be written in terms of the Kelvin functions ker and kei, as discussed by Krishna (1980):

$$U = cG \ker(x) + U_G, \qquad V = cG \ker(x) + V_G \tag{A2}$$

where $x$ is a normalized height $x \equiv 2\sqrt{z|f_c|/(\kappa u_{*0})}$ and $c$ is a constant. For $z \to z_0$ (and assuming $z_0 \ll u_{*0}/|f_c|$), the Kelvin functions can be expanded, and the solution can be written as:

$$U \approx -cG\left[\frac{1}{2}\ln\left(\frac{z_0|f_c|}{\kappa u_{*0}}\right) + \gamma_e\right] + U_G = 0, \qquad V \approx -cG\frac{\pi}{4} + V_G = 0 \tag{A3}$$

where $\gamma_e \approx 0.57721$ is the Euler-Mascheroni constant. We can set the geostrophic wind $G$ through the constant $c$:

$$c = -\left(\left[\frac{1}{2}\ln\left(\frac{z_0|f_c|}{\kappa u_{*0}}\right) + \gamma_e\right]^2 + \frac{\pi^2}{16}\right)^{-1/2}, \qquad U_G = cG\left[\frac{1}{2}\ln\left(\frac{z_0|f_c|}{\kappa u_{*0}}\right) + \gamma_e\right], \qquad V_G = cG\frac{\pi}{4} \tag{A4}$$

Note that Krishna (1980) chose $cG = -2u_{*0}/\kappa$ (so his $-c$ is five times the geostrophic drag coefficient $u_{*0}/G$ for $\kappa = 0.4$), which follows from the Neumann condition:

$$\frac{dU}{dz} = \frac{u_{*0}}{\kappa z} = -\frac{cG}{2z} \tag{A5}$$

by taking $d/dz$ of $U$ from Eq. (A2) for $z \to z_0$. As as consequence, the geostrophic wind becomes a dependent parameter. We prefer to keep the geostrophic wind as an independent parameter by using $c$ as defined in Eq. (A4). Then, the effective $u_{*0}$ is calculated as $u_{*0,\text{eff}} = cG\kappa/2$.

A GDL can be derived in form of Eq. (26) (using the Neumann conditions of Eq. (A5) and the constant $c$ from Eq. (A4)), where $A = -\ln(\kappa) + 2\gamma_e \approx 2.07$ and $B = \pi/2 \approx 1.57$, as also shown by Krishna (1980). The friction velocity in Eq. (A4) can now be calculated by solving the GDL for $u_{*0}/G$. Hence, the analytic solution of Ellison (1956) is only dependent on $\text{Ro}_0$.

The cross isobar angle (angle between the geostrophic wind direction and surface wind direction) can be written as a function of the geostrophic drag coefficient $u_{*0}/G$ and the Rossby number $\text{Ro}_0$ using Eq. (A4):

$$\theta_0 = \arctan\left(\frac{V_G}{U_G}\right) = \arctan\left(\frac{\pi/2}{2\gamma_e - \ln(\text{Ro}_0) - \ln(\kappa u_{*0}/G)}\right) \tag{A6}$$

where the GDL can be used to solve for $u_{*0}/G$.