# Peer review of "Rossby number similarity of atmospheric RANS using limited length scale turbulence closures extended to unstable stratification"

_Wind Energy Science, 2019_

## Referee Comment (RC1) · Javier Sanz Rodrigo (Referee) · 14 Nov 2019

General Assessment

Interesting paper extending the range of idealized ABL models to include more realistic scaling in unstable conditions in connection to wind energy design tools. I appreciate the effort of the authors in explaining the derivation of the extended model from original models dating back to Ekman (1905). This, in itself, makes the paper worthy of publication to understand a historical perspective on ABL modeling. The authors convincingly

demonstrate the scaling properties of the ABL limited-mixing-length model when using Rossby-based length scales which is convenient to reduce the dimensionality of ABL parameterization. The model has a good theoretical basis to provide a more realistic framework for design tools than traditional surface-layer models while it still struggles at reproducing real ABL profiles that are inherently transient and driven by non-uniform forcing. This is demonstrated with a series of validation cases.

There is some clarification to be made on the use or not of local-scaling in stable conditions and, in general, the use of u* vs G and how this could affect Rossby similarity. This is discussed in page 13 but maybe I should be motivated before the model derivation.

In the conclusions, I miss a more extended discussion about the applicability of this model for wind energy applications and challenges that will arise when dealing, for instance, with complex terrain or wake effects. Will the same length scales apply? Will there be additional length scales?

Major Remarks

In the derivation of the model, we end up having three Rossby numbers related to z_0, l_max (or l_ABL in stable conditions) and L_ length scales. All three use G/abs(fc) to come up with the non-dimensional number. I understand the convenience of using the same velocity scale for all three parameters but somehow it implies that there is global scaling for all stability conditions. Since we know that the stable ABL depends on local scaling (e.g. Nieuwstadt, JAS-14, 1984) through z/L I would like to confirm if this is already implicit in l_max through equation (13). If so, I would suggest that L is defined as the local Obukhov length when it is first introduced in equation (11). Otherwise, if L is the surface Obukhov length (as suggested in page 10-21) then you could expect some difficulties of the model to fit very stable profiles as it might be the case in Figure 8.

Along this local-scaling reasoning, I would find it more appropriate to use u* as a velocity scale in relation to l_ABL which dependes on z/L, with L being a local quantity. Actually, you can generalize l_ABL to unstable conditions by simply using the equivalent Dyer functional forms of the stability function phi_m(z/L). In this alternative formulation you would have z_0, l_max (Blackadar) and L as length scales with z_0 and l_max being global scales and L being local scale. I'm not sure if this would work out in terms of Rossby similarity, as you mention at the beginning of page 13.

Page 6, Equation (13): I would define l_ABL = l_max for z/L = 0 and l_ABL = [1/l_max + beta/(kappa*L)]^-1 and use l_ABL in the definition of Ro_l (21) since l_max is originally associated with Blackadar's mixing length for neutral conditions and you will use Ro_l for both neutral and stable conditions. Then, equation (21) would be based on l_ABL, and not l_max right?

Minor Remarks

Page 1-15: "Such a model should be simple enough to be applicable in the wind energy industry" It sounds a bit like industry could not handle complex models. Maybe, "... to be efficiently used in design tools".

For completeness, I think you should mention in the introduction other hypothesis that apply to the model even if some of them are mentioned in the derivation later on: dry-atmosphere, no mesoscale advection, no vertical wind speed, etc.

2-16: Even if you mention them later, please provide references to the "Blackadar-type" models to provide a more meaningful introduction of the type of models that you try to improve from.

5-18: Consider changing the title of the section to "Limited mixing-length model in stable conditions" since the objective is to define l_ABL which, in this formulation, do not include unstable conditions

7-9: For clarity, "... all have the same coefficient: C_epsilon,2."

8-12: A rough-wall boundary condition

9-7: L was used before to denote the Obukhov length. Although I think it is clear from the context, I would rather use U* and L* or anything else to denote that these are generic velocity and length scales, not to be confused with U and L elsewhere in the paper.

9-10: "Ro_l is analogous to the reciprocal of a dimensionless boundary layer depth" Maybe you could add the dimensionless boundary layer depth for clarity. This will help when you interpret equation (23) as a ratio of $z_i/z_0$ (page 10-1)

Equation (22): Following previous comment, you may consider naming $l_b = G/abs(fc)$ as a "master" length scale (not sure about the most appropriate name) since this is present throughout the paper and you also use this to plot non-dimensional height in Figures that follow.

16-9: Why the blue line and not the yellow line with A= 2 and B =4.42 being somehow closer to A = 1.8 and B = 4.5 (Troen and Petersen)?

---

## Referee Comment (RC2) · Anonymous Referee #2 · 10 Dec 2019

Review of the manuscript:

Rossby number similarity of atmospheric RANS using limited lengthscale turbulence closures extended to unstable stratification Maarten Paul van der Laan, Mark Kelly, Rogier Floors, and Alfredo Peña

General comments:

In this paper, a one-dimensional RANS model is used to simulate idealised ABL profiles and shows that, for neutral and stable conditions, the Blackadar limited length

scale model, and the limited length-scale k-e model of Apsley and Castro produce profiles that can be described by two Rossby numbers: one using z0 as the characteristic lengthscale, the other one using the maximum turbulence lengthscale defining the model. For unstable conditions, an extension is proposed to the limited lengthscale k-e model, which produce profiles characterised by a third Rossby number depending on the Obukhov lengthscale. The model derived drag law parameters A & B fit well within the range of observed typical A & B values. The model's ability to reproduce measured ABL profiles is tested with a varying level of success. The investigation is interesting, and potentially useful to provide inflow boundary conditions to a 3D model from a library of precalculated profiles. The application of the derived profiles within a 3D model hasn't been attempted yet in this contribution. The paper is generally well written, although some points/comments should be addressed as listed below.

Specific comments:

p.1, line 16. What does 'simple enough to be applicable in the wind energy industry' mean? So it can be used for wind turbine design? So it can be used as inflow b.c. for a flow model?

P.1, line 24-25. 'These turbulence models can simulate stable and neutral ABLs without the necessity of a temperature equation and a momentum source term of buoyancy. In other words, all temperature effects are represented by the turbulence model'. This is only true when looking at horizontally homogeneous flows (i.e. the 1D flows modelled in this paper). Once terrain, coastal discontinuities, or even an offshore wind farm is perturbing the flow, gravity waves can develop, which have the potential to affect the wind speed distribution at hub height. And to capture these you need the buoyancy source term in the momentum equation. Please elaborate on these. Your statement as it is can be misleading and let the reader assume that they can generally ignore the buoyancy in the momentum equation.

p.6 equation 13. We have two l_max here, l_max and l_max_eff. Am I right in understanding that when later on there is a reference to l_max it is in fact l_max_eff ? i.e. the Rossby number as a function of lmax or the fitted values of lmax in Table 2 are referring to values of lmax,eff? Is eq 13 used at all in your model? Probably worth clarifying. . . especially if I have misunderstood.

p.6. extended turbulence model for unstable flows: am I right in understanding that the buoyancy term B added to the k-e equations is only added for unstable flows. i.e. when you model neutral or stable cases listed in Table 2, you only use the original length scale limited model of Apsley and Castro?

p.8. Numerical set up. How is convergence defined? I'm trying to understand how the overall momentum balance is achieved. We have friction at the ground, so a momentum sink, no wind speed gradient at the top (so no shear driven flow). As friction is reduced or increased I'd expect the boundary layer height to reduce or grow vs time. Do you judge convergence based on the boundary layer growth? Do you prescribe a pressure gradient in the flow direction? Worth elaborating?

p.8. Numerical et up. Boundary condition at the ground: is it using the neutral formulation even when modelling stable or unstable cases? Did you try changing the closure at the wall using stability dependent closures?

p.9. line 4. I find the sentence 'We find that if both lmax and z0 are proportional to . . .' a bit of a back to front way to introduce Rossby number similarity. My first reaction when reading this was that z0 is usually an input parameter that depends on the ground conditions, therefore why should it be proportional to G/fc. Worth rewriting?

P9. Line 6. The general definition of Ro as a function of U and L where U and L are characteristic wind speed and velocity scales is fine on its own. But in the current context, where the symbol L has also been used for the Obukhov length, the use of L for a general lengthscale is a bit unfortunate. Especially since you proceed using L, the Obukhov length, when later defining RO_L-. I would suggest using a different symbol for L here, may be using a different font.

p.14, equation 25. I know it's common to use the A and B notation for the 'constants' in the GDL, but it's unfortunate that B was also the symbol used for the buoyancy term earlier on. I'd suggest avoiding the use of the same symbol for both.

p.16, line 8. The fact that the A and B parameters in the DGL are function of the stability (via lmax or L) is not new. While not necessarily formulated as a function of the Rossby numbers used in the current publication, their dependence on the Obukhov length and on the Brundt Vaysala frequency has been discussed quite a while ago. See e.g. Landberg for dependence in mu (i.e. Obukhov length)

Landberg, L., 1994, 'Short term prediction of local wind conditions', Risoe National Laboratory, Roskilde, Denmark

or Zilitinkevich (1989) already referred to. This would be worth including in the discussion.

p.18, Figure 8. Plot of the turbulence lengthscale. What is happening with the plot of the very stable case at the top of the ABL? Is the black line really the solution of the 1D CFD?

p.21. I find the discussion somewhat lacking on the fact that the profiles obtained by fitting both G, lmax were well captured, while those where only lmax was varied were not so well captured. Could it be because the role of lmax is to reflect the ABL height, while at the same time be accounting for surface stability effects? Should there be a third lengthscale also entering the definition of the stable profiles so that the role of limiting effects at the surface (via L) and limiting effects at the top of the boundary layer (via the Brundt Vaysala frequency) can be treated independently? This might provide the additional degree of freedom that the model seems to require to fit the measured profiles. (degree of freedom which was provided by allowing the model to fit G). This sort of dependency was proposed by Zilitinkevich and Mironov (1996) and it's use suggested in Zilitinkevich et al (1996). Worth discussing?

Zilitinkevich S.S., Mironov, D.V., 1996, 'A Multi-Limit Formulation For The Equilibrium Depth Of A Stably Stratified Boundary Layer', Bound. Layer Meteorol., 81, pp 325-351.
Zilitinkevich S.S., Johansson P.-E., Mironov D.V., Baklanov A., in press, 'An Analytical Similarity Theory Model For Wind Profile And Resistance Law In Stably Stratified Planetary Boundary Layers', J. Wind. Eng. Industr. Aerodyn, 74-76 (1998) 209-218.

P.21, line 25 '...dependence upon two Rossby numbers, which correspond to the roughness length and the maximum turbulence length scale' should be rewritten. 'correspond' is not exactly appropriate, may be use 'defined from' instead. p.22. The Obukhov length is dimensional, while the Rossby number is not. So the sentence '...the Obukhov length, which can also be written as a third Rossby number' should be changed. Likely something along the lines of 'The Obukhov length can be used to define a third Rossby number'.

p.22. The conclusion feels like it was hastily written. When I first read the abstract, introduction and conclusion, I could not quite understand what various parts of the conclusion referred to. I feel the conclusions should be improved, so that they can provide a clearer summary of what was done, so they can stand on their own without having to read through the whole article. For example, the content of the sentence 'A model validation of the full ABL for a stable, a neutral and an unstable case is performed, with less success for the non-neutral cases.' could be explained a bit more. i.e. what is meant by a full validation? Also, the results of the validation could be detailed a bit more than the rather succint 'with less success for the non-neutral case'.

Technical corrections:

p.2, line 31. Negative sign missing in front of u'w' and v'w'.

p.2 eq (1). Should be a positive sign in front of the diffusion terms

p.8, line 12. Comma instead of full stop after 'at the ground'

p.14, line 6. Should be proportional to minus a log10(Ro_l) (negative slope in Fig 5)

p.24. Zilitinkevich reference is missing the journal.

---

## Author Comment (AC1) · 17 Jan 2020

**Reply to reviewers**

January 17, 2020

We would like to thank the two reviewers for their detailed feedback and suggestions to improve the article. In the next sections, the reviewers comments are copied and answered per comment (blue color). An additional document is provided that highlights all modifications with respect to the initial submitted version.

**Reviewer 1 (Javier Sanz Rodrigo)**

**General Assessment**

Interesting paper extending the range of idealized ABL models to include more realistic scaling in unstable conditions in connection to wind energy design tools. I appreciate the effort of the authors in explaining the derivation of the extended model from original models dating back to Ekman (1905). This, in itself, makes the paper worthy of publication to understand a historical perspective on ABL modeling. The authors convincingly demonstrate the scaling properties of the ABL limited-mixing-length model when using Rossby-based length scales which is convenient to reduce the dimensionality of ABL parameterization. The model has a good theoretical basis to provide a more realistic framework for design tools than traditional surface-layer models while it still struggles at reproducing real ABL profiles that are inherently transient and driven by non-uniform forcing. This is demonstrated with a series of validation cases.

There is some clarification to be made on the use or not of local-scaling in stable conditions and, in general, the use of u* vs G and how this could affect Rossby similarity. This is discussed in page 13 but maybe I should be motivated before the model derivation.

In the conclusions, I miss a more extended discussion about the applicability of this model for wind energy applications and challenges that will arise when dealing, for instance, with complex terrain or wake effects. Will the same length scales apply? Will there be additional length scales?

**Major Remarks**

1. In the derivation of the model, we end up having three Rossby numbers related to z_0, l_max (or l_ABL in stable conditions) and L_ length scales. All three use G/abs(fc) to come up with the non-dimensional number. I understand the convenience of using the same velocity scale for all three parameters but somehow it implies that there is global scaling for all stability conditions. Since we know that the stable ABL depends on local scaling (e.g. Nieuwstadt, JAS-14, 1984) through z/L I would like to confirm if this is already implicit in l_max through equation (13). If so, I would suggest that L is defined as the local Obukhov length when it is first introduced in equation (11). Otherwise, if L is the surface Obukhov length (as suggested in page 10-21) then you could expect some difficulties of the model to fit very stable profiles as it might be the case in Figure 8.

It's nice to consider very stable cases and local similarity. Recall that MOST is a surface-layer theory in stable conditions, for common values of $1/L$; local scaling arises at larger values of $z/L$ (see e.g. Wyngaard, 2010 text). We use the original definition of $L$ (nonlocal); this is consistent with the modelling, which includes $L$ prescribed via *phi*-functions—which are ASL-based. The very stable case in Fig. 8 has $z/L < 1$ ($L^{-1} = 0.036\,\mathrm{m}^{-1}$), not in the range of local-similarity (valid for $z/L \gg 1$; local

MOST can arise in the RANS equations, if buoyancy is implemented differently, e.g. via $d\theta/dt$ equation and source terms in $\{w, k, \varepsilon\}$.)

2. Along this local-scaling reasoning, I would find it more appropriate to use u* as a velocity scale in relation to l_ABL which dependes on z/L, with L being a local quantity. Actually, you can generalize l_ABL to unstable conditions by simply using the equivalent Dyer functional forms of the stability function phi_m(z/L). In this alternative formulation you would have z_0, l_max (Blackadar) and L as length scales with z_0 and l_max being global scales and L being local scale. I'm not sure if this would work out in terms of Rossby similarity, as you mention at the beginning of page 13.

This is a good point. We (the authors) had discussions about the use of either $G$ versus $u_{*0}$, and purposefully decided to use $G$. It is more convenient to use $G$ instead of $u_{*0}$, because $G$ is a model input, while $u_{*0}$ is a model result. In addition, $u_{*0}$ in practice can become a function of height (more so for shallower ABLs, despite being assumed constant); i.e., the height at which $u_{*0}$ is diagnosed then becomes an extra 'input'. However, it is possible to obtain a Rossby similarity based on ideal (constant, ASL value) $u_{*0}$. We have now added an Appendix (Appendix B) showing exactly this, which makes the Rossby similarity more general—since the choice of velocity scale does not affect the similarity. We have also defined three alternative Rossby numbers based on $u_{*0}$, which can be written in terms of the Rossby numbers based on $G$ and the geostrophic drag coefficient $u_{*0}/G$:

$$\mathrm{Ro}_\ell^* \equiv \frac{u_{*0}}{|f_c|\ell_{\max}} = \frac{u_{*0}}{G}\mathrm{Ro}_\ell, \qquad \mathrm{Ro}_0^* \equiv \frac{u_{*0}}{|f_c|z_0} = \frac{u_{*0}}{G}\mathrm{Ro}_0, \qquad \mathrm{Ro}_{L_-}^* \equiv \frac{-u_{*0}}{|f_c|L} = \frac{u_{*0}}{G}\mathrm{Ro}_{L_-}$$

The geostrophic drag coefficient $u_{*0}/G$ as plotted in Fig. 6 can thus be transformed as an explicit relation of $\mathrm{Ro}^*$, $\mathrm{Ro}_0^*$ and $\mathrm{Ro}_{L_-}^*$. This information gives the value $G$ needed to obtain a desired $u_{*0}$ for Rossby numbers based on $u_{*0}$. Figure 1 below depicts the Rossby similarity based on $u_{*0}$ using the limited length scale $k$-$\varepsilon$ model for neutral conditions ($\mathrm{Ro}_{L_-}^* = 0$); but we note it also works for $\mathrm{Ro}_{L_-}^* \neq 0$ and using the limited mixing-length model instead of the limited length scale $k$-$\varepsilon$ model.

[Figure]

| $\mathrm{Ro}_0^*, \mathrm{Ro}_\ell^*$ | $f_c[\mathrm{s}^{-1}], u_{*0}[\mathrm{ms}^{-1}]$ | $f_c[\mathrm{s}^{-1}], u_{*0}[\mathrm{ms}^{-1}]$ | $f_c[\mathrm{s}^{-1}], u_{*0}[\mathrm{ms}^{-1}]$ | $f_c[\mathrm{s}^{-1}], u_{*0}[\mathrm{ms}^{-1}]$ |
|---|---|---|---|---|
| $10^5, 10^2$: | $5 \times 10^{-5}, 0.2$ | $5 \times 10^{-5}, 0.4$ | $10^{-4}, 0.2$ | $10^{-4}, 0.4$ |
| $10^5, 10^4$: | $5 \times 10^{-5}, 0.2$ | $5 \times 10^{-5}, 0.4$ | $10^{-4}, 0.2$ | $10^{-4}, 0.4$ |
| $10^8, 10^2$: | $5 \times 10^{-5}, 0.2$ | $5 \times 10^{-5}, 0.4$ | $10^{-4}, 0.2$ | $10^{-4}, 0.4$ |
| $10^8, 10^4$: | $5 \times 10^{-5}, 0.2$ | $5 \times 10^{-5}, 0.4$ | $10^{-4}, 0.2$ | $10^{-4}, 0.4$ |

Figure 1: [In response to major remark #2] Rossby number similarity of the limited length scale $k$-$\varepsilon$ model using the friction velocity as the velocity scale, for neutral conditions ($\mathrm{Ro}_{L_-} = 0$.)

Regarding use of the unstable $\phi_m$ function to extend $\ell_{\max}$ for unstable conditions, this is exactly what we do for the mixing-length model, where the turbulence length scale is prescribed analytically (see Section 3.1 and Eqns. 9-11). This does not work for the limited length scale $k$-$\varepsilon$ model of Apsley and Castro, because the turbulence length scale is always smaller than $\kappa z$, no matter what the value of $\ell_{\max}$ is (except for a possible overshoot near the ABL height). To include length scales that are larger than $\kappa z$, one needs to include an additional model behavior, i.e. buoyancy source terms in the turbulence transport equations as given in Section 3.2.

We remind that local scaling is not valid for unstable conditions (though mixed-layer scaling arises in the unstable ABL), again consistent with our $L$ being surface-based.

3. Page 6, Equation (13): I would define l_ABL = l_max for z/L = 0 and l_ABL = [1/l_max + beta/(kappa*L)]^-1 and use l_ABL in the definition of Ro_l (21) since l_max is originally associated with Blackadar's mixing length for neutral conditions and you will use Ro_l for both neutral and stable conditions. Then, equation (21) would be based on l_ABL, and not l_max right?

We understand that one could parametrize the (stable) turbulence length scale of the ABL, $\ell_{\mathrm{ABL}}$, using a stable $\phi_m$ function, as shown by Apsley and Castro (1997). However, one could also think of a more general $\ell_{\mathrm{ABL}}$ by not only using $\ell_{\max}$ and $\phi_m$, but also including additional length scales that could influence $\ell_{\mathrm{ABL}}$, e.g., entrainment. This is something we plan to look at in future work. In addition, your second (stable) definition of $\ell_{\mathrm{ABL}}$, can be rewritten as an effective $\ell_{\max}$, as shown in Eq. (13), hence a stable $L$ is not an independent parameter and therefore, we use a Rossby number based on $\ell_{\mathrm{ABL}} = \ell_{\max}$. This is not the case for unstable conditions, where the $\phi_m$ function is not linear in $z/L$, and therefore we end up with third independent parameter in form of an unstable $L$ or its associated Rossby number. We have added a small clarification about the effective $\ell_{\max}$ at the end of Section 3.1 as a response to the second reviewer.

**Minor Remarks**

1. Page 1-15: "Such a model should be simple enough to be applicable in the wind energy industry" It sounds a bit like industry could not handle complex models. Maybe, "... to be efficiently used in design tools".
   *You are right, this sentence can be interpreted in that way. We mean that the wind energy industry can benefit more from simple and fast models that are often more easy to be adapted in a chain of design tools compared to high fidelity models. We have changed the sentence to: Such a model should be simple enough to efficiently improve the chain of design tools used by the wind energy industry.*

2. For completeness, I think you should mention in the introduction other hypothesis that apply to the model even if some of them are mentioned in the derivation later on: dry-atmosphere, no mesoscale advection, no vertical wind speed, etc.
   *We have added here we exclude effects of flow inhomogeneity and nonstationarity, which are typically considered by mesoscale and three-dimensional time-varying model in: Idealized, steady-state models can represent long-term averaged velocity and turbulence profiles of the real ABL, including the effects of Coriolis, atmospheric stability, capping inversion, homogeneous surface roughness and flat terrain; here we exclude effects of flow inhomogeneity and nonstationarity, which are typically considered by mesoscale and three-dimensional time-varying models. We think that all the other assumptions are stated clear enough in the introduction.*

3. 2-16: Even if you mention them later, please provide references to the "Blackadar-type" models to provide a more meaningful introduction of the type of models that you try to improve from.
   *We now refer to Blackadar (1962). In addition, we have rephrased a sentence in Section 3.2 to: In stable conditions, $\phi_m = 1 + \beta z/L$, so the resulting turbulence length can also be rewritten in the form of Eqs. (4) and (9),...*

4. 5-18: Consider changing the title of the section to "Limited mixing-length model in stable conditions" since the objective is to define l_ABL which, in this formulation, do not include unstable conditions
   *In this Section, we start with the original models, suited for stable and neutral conditions, and then extend it to unstable conditions. Hence adding "for stable conditions" to the subsection title does not make sense. We have decided to keep the current title.*

5. 7-9: For clarity, "... all have the same coefficient: C_epsilon,2."
   *Adapted.*

6. 8-12: A rough-wall boundary condition
   *Corrected.*

7. 9-7: L was used before to denote the Obukhov length. Although I think it is clear from the context, I would rather use U* and L* or anything else to denote that these are generic velocity and length scales, not to be confused with U and L elsewhere in the paper.
   *This is a good point. We have now used $\mathcal{U}$ and $\mathcal{L}$ as the characteristic velocity and length scales.*

8. 9-10: "Ro_l is analogous to the reciprocal of a dimensionless boundary layer depth" Maybe you could add the dimensionless boundary layer depth for clarity. This will help when you interpret equation (23) as a ratio of z_i/z_0 (page 10-1)
   *We have added $z_i f_c/u_{*0}$.*

9. Equation (22): Following previous comment, you may consider naming l_b = G/abs(fc) as a "master" length scale (not sure about the most appropriate name) since this is present throughout the paper and you also use this to plot non-dimensional height in Figures that follow.
   *Such a master length scale could be used, but we prefer to stick with $G$ and $f_c$ in the plots because they are well-known quantities.*

10. 16-9: Why the blue line and not the yellow line with A = 2 and B = 4.42 being somehow closer to A = 1.8 and B = 4.5 (Troen and Petersen)?
    *The simple 'blue' statement arose from an earlier version of the plot. We now correct; For moderate*

*roughness lengths over land, the measured values tabulated by Hess and Garratt (2002) generally fall between the blue and yellow lines for neutral conditions, which are consistent with the typically used values in wind energy are $A = 1.8$ and $B = 4.5$ (e.g. Troen and Petersen, 1989).*

**Reviewer 2 (Anonymous)**

In this paper, a one-dimensional RANS model is used to simulate idealised ABL profiles and shows that, for neutral and stable conditions, the Blackadar limited length scale model, and the limited length-scale k-e model of Apsley and Castro produce profiles that can be described by two Rossby numbers: one using z0 as the characteristic lengthscale, the other one using the maximum turbulence lengthscale defining the model. For unstable conditions, an extension is proposed to the limited lengthscale k-e model, which produce profiles characterised by a third Rossby number depending on the Obukhov lengthscale. The model derived drag law parameters A B fit well within the range of observed typical A B values. The model's ability to reproduce measured ABL profiles is tested with a varying level of success. The investigation is interesting, and potentially useful to provide inflow boundary conditions to a 3D model from a library of precalculated profiles. The application of the derived profiles within a 3D model hasn't been attempted yet in this contribution. The paper is generally well written, although some points/comments should be addressed as listed below.

**Specific comments:**

1. p.1, line 16. What does 'simple enough to be applicable in the wind energy industry' mean? So it can be used for wind turbine design? So it can be used as inflow b.c. for a flow model?
   We mean that the wind energy industry can benefit more from simple and fast models that are often more easy to be adapted in a chain of design tools compared to high fidelity models. We have changed the sentence to: *Such a model should be simple enough to efficiently improve the chain of design tools used by the wind energy industry.*

2. P.1, line 24-25. 'These turbulence models can simulate stable and neutral ABLs without the necessity of a temperature equation and a momentum source term of buoyancy. In other words, all temperature effects are represented by the turbulence model'. This is only true when looking at horizontally homogeneous flows (i.e. the 1D flows modelled in this paper). Once terrain, coastal discontinuities, or even an offshore wind farm is perturbing the flow, gravity waves can develop, which have the potential to affect the wind speed distribution at hub height. And to capture these you need the buoyancy source term in the momentum equation. Please elaborate on these. Your statement as it is can be misleading and let the reader assume that they can generally ignore the buoyancy in the momentum equation.
   This is a valid point. One may need a buoyancy source term in the momentum equation to accurately model the effects of stability in complex terrain and wind farms. However, we don't know how much the simplification (of modeling the effect of stability without a momentum source term of buoyancy) would affect the flow. We have added **one dimensional** in the introduction: *These turbulence models can simulate one dimensional stable and neutral ABLs without the necessity of a temperature equation and a momentum source term of buoyancy.* In addition, we have added a small discussion on this topic at the end of the conclusion: *In addition, the effects of length scale limitation and neglecting the buoyancy force in the momentum equation need to be quantified for three-dimensional RANS simulations of complex terrain and wind farms.*

3. p.6 equation 13. We have two l_max here, l_max and l_max_eff. Am I right in understanding that when later on there is a reference to l_max it is in fact l_max_eff ? i.e. the Rossby number as a function of lmax or the fitted values of lmax in Table 2 are referring to values of lmax,eff? Is eq 13 used at all in your model? Probably worth clarifying... especially if I have misunderstood.
   Equation (13) is just to show that it does not make sense to define a surface layer turbulence length scale, $\ell_{\mathrm{ASL}}$, based on MOST for stable conditions, because one would obtain the original turbulence length scale expression of Blackadar (1962). We have added clarifying text: *Thus we can simply use the original length scale model of Blackadar (1962) for stable and neutral conditions; the stable $\phi_m$ function simply informs the selection of $\ell_{\mathrm{max,eff}}$, following Eq. (13).*

4. p.6. extended turbulence model for unstable flows: am I right in understanding that the buoyancy term B added to the k-e equations is only added for unstable flows. i.e. when you model neutral or stable cases listed in Table 2, you only use the original length scale limited model of Apsley and Castro?

   That is correct. One could add the buoyancy terms for neutral conditions, but they would be zero because then we have $1/L = 0$.

5. p.8. Numerical set up. How is convergence defined? I'm trying to understand how the overall momentum balance is achieved. We have friction at the ground, so a momentum sink, no wind speed gradient at the top (so no shear driven flow). As friction is reduced or increased I'd expect the boundary layer height to reduce or grow vs time. Do you judge convergence based on the boundary layer growth? Do you prescribe a pressure gradient in the flow direction? Worth elaborating?

   The flow is driven by a set geostrophic wind speed $G$, which represents a constant prescibed pressure gradient, i.e, $\partial P/\partial x = \rho f_c V_G$ and $\partial P/\partial y = -\rho f_c U_G$. A balance between the pressure gradient, Coriolis force and the turbulence generated from the ground is achieved when the convergence has been achieved. The convergence is defined as the equation residual normalized by the initial guess, which is a uniform flow with the $G$ as the wind speed. We have added to Section 4.2: *The flow is driven by a constant pressure gradient using a prescribed constant geostrophic wind speed. The initial wind speed is set to the geostrophic wind speed. During the solving procedure the ABL depth grows from the ground until convergence is achieved, which occurs when the growth of the ABL depth is negligible because a balance of the prescribed pressure gradient, Coriolis forces and turbulence stresses is obtained..*

6. p.8. Numerical set up. Boundary condition at the ground: is it using the neutral formulation even when modelling stable or unstable cases? Did you try changing the closure at the wall using stability dependent closures?

   We use the same rough wall boundary conditions in neutral and non-neutral conditions. This is because the flow near the wall is following a neutral logarithmic profile, also in non-neutral conditions. Our first cell height is small enough (0.1 m) to be in this region and we do not need to extend the boundary conditions for non-neutral conditions. One could also see from the wind speed profile based on MOST, that one obtains the neutral logarithmic profile near the wall since $z/L \to 0$.

7. p.9. line 4. I find the sentence 'We find that if both lmax and z0 are proportional to ...' a bit of a back to front way to introduce Rossby number similarity. My first reaction when reading this was that z0 is usually an input parameter that depends on the ground conditions, therefore why should it be proportional to G/fc. Worth rewriting?

   We understand that this is somewhat confusing. We have now rewritten the first part of Section 5, where we have added a full derivation of the three Rossby numbers by introducing normalized variables in the momentum equation.

8. P9. Line 6. The general definition of Ro as a function of U and L where U and L are characteristic wind speed and velocity scales is fine on its own. But in the current context, where the symbol L has also been used for the Obukhov length, the use of L for a general lengthscale is a bit unfortunate. Especially since you proceed using L, the Obukhov length, when later defining RO_L-. I would suggest using a different symbol for L here, may be using a different font.

   This is a good point. We now used $\mathcal{U}$ and $\mathcal{L}$.

9. p.14, equation 25. I know it's common to use the A and B notation for the 'constants' in the GDL, but it's unfortunate that B was also the symbol used for the buoyancy term earlier on. I'd suggest avoiding the use of the same symbol for both.

   This is also good point. We have changed the Buoyancy variable $B$ to $\mathcal{B}$.

10. p.16, line 8. The fact that the A and B parameters in the DGL are function of the stability (via lmax or L) is not new. While not necessarily formulated as a function of the Rossby numbers used in the current publication, their dependence on the Obukhov length and on the Brundt Vaysala frequency has been discussed quite a while ago. See e.g. Landberg for dependence in mu (i.e. Obukhov length) Landberg, L., 1994, 'Short term prediction of local wind conditions', Risoe National Laboratory, Roskilde,

Denmark or Zilitinkevich (1989) already referred to. This would be worth including in the discussion. *We added: This is not a surprising result, since many authors showed that A and B are dependent on atmospheric stability (see, e.g., Arya 1975; Zilitinkevich 1989; Landberg 1994).*

11. p.18, Figure 8. Plot of the turbulence lengthscale. What is happening with the plot of the very stable case at the top of the ABL? Is the black line really the solution of the 1D CFD?
We obviously forgot to discuss this result. The simulated very stable case has an ABL depth around 100 m, see Fig. 8a. When the gradient $dU/dz$ is taken around the ABL depth, we get $dU/dz \to 0$ so that $\hat{\ell} = u_{*0}/(dU/dz) \to \infty$, which results in a spike in Fig 8b. We have added some clarification in the text: *... and the model predicts an ABL depth of about 100 m, which results in a spike in $\hat{\ell}$, since $dU/dz$ is zero around the ABL depth.*

12. p.21. I find the discussion somewhat lacking on the fact that the profiles obtained by fitting both G, lmax were well captured, while those where only lmax was varied were not so well captured. Could it be because the role of lmax is to reflect the ABL height, while at the same time be accounting for surface stability effects? Should there be a third lengthscale also entering the definition of the stable profiles so that the role of limiting effects at the surface (via L) and limiting effects at the top of the boundary layer (via the Brundt Vaysala frequency) can be treated independently? This might provide the additional degree of freedom that the model seems to require to fit the measured profiles. (degree of freedom which was provided by allowing the model to fit G). This sort of dependency was proposed by Zilitinkevich and Mironov (1996) and it's use suggested in Zilitinkevich et al (1996). Worth discussing? Zilitinkevich S.S., Mironov, D.V., 1996, 'A Multi-Limit Formulation For The Equilibrium Depth Of A Stably Stratified Boundary Layer', Bound. Layer Meteorol., 81, pp 325-351. Zilitinkevich S.S., Johansson P.-E., Mironov D.V., Baklanov A., in press, 'An Analytical Similarity Theory Model For Wind Profile And Resistance Law In Stably Stratified Planetary Boundary Layers', J. Wind. Eng. Industr. Aerodyn, 74-76 (1998) 209-218.
This is an interesting point, which we had discussed in previous months; we have plans to extend the $k$-$\varepsilon$ model further to include effects of an inversion strength (via e.g. $N_{BV}$), as this is very important parameter for wind farm simulations subjected to atmospheric inflow. However, we chose and prefer to save this discussion for an upcoming article, because we would need to add a significant part, to an already lengthy article.

13. P.21, line 25 '...dependence upon two Rossby numbers, which correspond to the roughness length and the maximum turbulence length scale' should be rewritten. 'correspond' is not exactly appropriate, may be use 'defined from' instead.
Adapted.

14. p.22. The Obukhov length is dimensional, while the Rossby number is not. So the sentence '...the Obukhov length, which can also be written as a third Rossby number' should be changed. Likely something along the lines of 'The Obukhov length can be used to define a third Rossby number'.
Adapted.

15. p.22. The conclusion feels like it was hastily written. When I first read the abstract, introduction and conclusion, I could not quite understand what various parts of the conclusion referred to. I feel the conclusions should be improved, so that they can provide a clearer summary of what was done, so they can stand on their own without having to read through the whole article. For example, the content of the sentence 'A model validation of the full ABL for a stable, a neutral and an unstable case is performed, with less success for the non-neutral cases.' could be explained a bit more. i.e. what is meant by a full validation? Also, the results of the validation could be detailed a bit more than the rather succint 'with less success for the non-neutral case'.
In general we like a short conclusion, but we understand that we could improve it based on your suggestions. We have remove the word *full* from *full ABL* because we just mean a boundary layer profile that does not only corresponds to a surface layer. In addition, we have added *In the very stable case, the measured wind veer of $50°$ was larger than the maximum wind veer of $45°$ that the $k$-$\varepsilon$ model can simulate. In addition, the very unstable case was characterized by non-stationary conditions, which are difficult to capture with a RANS model.*

**Technical corrections:**

1. p.2, line 31. Negative sign missing in front of u'w' and v'w'.
   Corrected.

2. p.2 eq (1). Should be a positive sign in front of the diffusion terms
   Corrected.

3. p.8, line 12. Comma instead of full stop after 'at the ground'
   Corrected.

4. p.14, line 6. Should be proportional to minus a $\log 10(\text{Ro\_l})$ (negative slope in Fig 5)
   Corrected.

5. p.24. Zilitinkevich reference is missing the journal.
   Corrected.

[revised manuscript text omitted]